# The structural basis for $2' - 5'/3' - 5'$-cGAMP synthesis by cGAS

Shuai Wu [1], Sandra B. Gabelli [1,2,3,5] & Jungsan Sohn [1,2,4]✉

cGAS activates innate immune responses against cytosolic double-stranded DNA. Here, by determining crystal structures of cGAS at various reaction stages, we report a unifying catalytic mechanism. apo-cGAS assumes an array of inactive conformations and binds NTPs nonproductively. Dimerization-coupled double-stranded DNA-binding then affixes the active site into a rigid lock for productive metal•substrate binding. A web-like network of pro-tein•NTP, intra-NTP, and inter-NTP interactions ensures the stepwise synthesis of $2'-5'/3'-5'$-linked cGAMP while discriminating against noncognate NTPs and off-pathway intermediates. One divalent metal is sufficient for productive substrate binding, and capturing the second divalent metal is tightly coupled to nucleotide and linkage specificities, a process which manganese is preferred over magnesium by 100-fold. Additionally, we elucidate how mouse cGAS achieves more stringent NTP and linkage specificities than human cGAS. Together, our results reveal that an adaptable, yet precise lock-and-key-like mechanism underpins cGAS catalysis.

Cyclic GMP-AMP (cGAMP) synthase (cGAS) is essential for the host defense against cytosolic double-stranded (ds)DNA arising from various maladies (e.g., pathogen invasion, ionizing irradiation, and genotoxic chemicals)[1–3]. Upon directly binding to and dimerizing on dsDNA, cGAS cyclizes ATP and GTP into $2'-5'/3'-5'$-linked cGAMP, a unique metazoan second messenger for initiating type-I interferon (IFN-I)-mediated inflammatory responses (Fig. 1A)[1,4–7]. cGAS is central to antitumor immunity, host defense against an array of pathogens, and regulating autoimmunity[2,3,8].

It is increasingly appreciated that cGAS-like enzymes have an ancestral origin and cGAMPs are widely employed as second messengers for both prokaryotic and eukaryotic innate immune pathways[9]. However, only metazoan enzymes generate the $2'-5'/3'-5'$-linkage combination[4,5,9,10], and currently, the mechanisms by which cGAS specifically generates this uniquely linked cyclic dinucleotide remain poorly understood. Persisting questions include: how cGAS specifically recognizes ATP and GTP at each substrate binding pocket and coordinates its signature $2'-5'$ linkage formation; how the same substrate binding sites then switch their nucleotide specificity and precisely

position the GTP-AMP (pppGpA) intermediate for the second $3'-5'$-linkage formation while barring the cyclization of other off-pathway dinucleotides; how dsDNA and dimerization regulate these processes, how it utilizes different divalent metals; and finally, how active site reactivity and promiscuity are regulated across different mammalian species. Here, we resolve these numerous fundamental mechanistic questions in innate immunology and present a unifying catalytic mechanism of cGAS.

We find that apo-cGAS assumes an array of inactive conformations and binds ATP/GTP nonproductively without involving divalent metals, which is then affixed into the catalytically competent conformation by dimerization-coupled dsDNA binding. We also delineate how web-like network of protein•NTP, intra-NTP, and inter-NTP inter-actions underpin the substrate-dependent linkage specificity. One $Mg^{2+}$ is sufficient for productive substrate binding and $Mn^{2+}$ is preferred as the second catalytic metal without involving an inverted intermediate. In the cyclization step, the adenosine of the pppGpA intermediate binds ~30° rotated compared to the guanine of the GTP substrate to precisely position the 3′-OH for the second linkage

[1]Department of Biophysics and Biophysical Chemistry, Johns Hopkins University School of Medicine, Baltimore, MD, USA. [2]Department of Oncology, Johns Hopkins University School of Medicine, Baltimore, MD, USA. [3]Department of Medicine, Johns Hopkins University School of Medicine, Baltimore, MD, USA. [4]Division of Rheumatology, Johns Hopkins University School of Medicine, Baltimore, MD, USA. [5]Present address: Discovery Chemistry, Merck Laboratories, West Point, PA, USA. ✉e-mail: jsohn@jhmi.edu

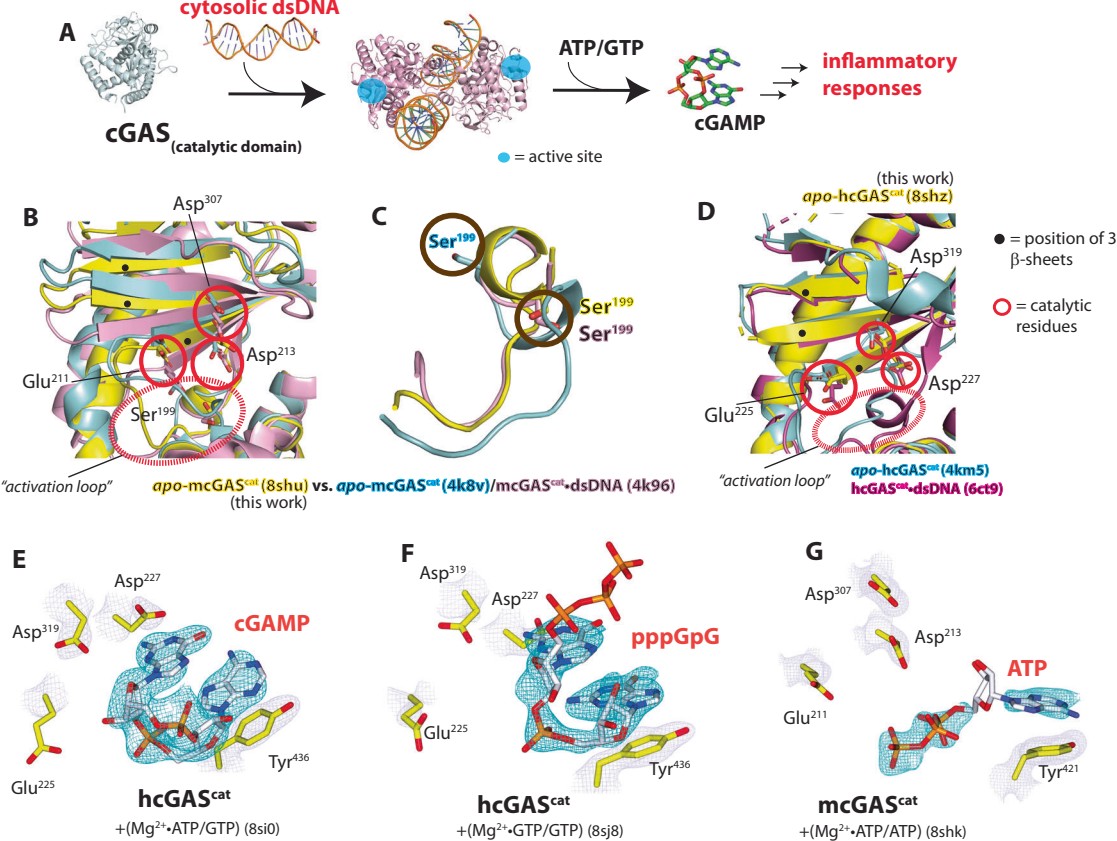

**Fig. 1 | The activation of cGAS entails a disorder-to-order transition. A** Cartoon describing the activation of cGAS. **B, C** An overlay of dsDNA-free and dsDNA-bound wild-type (WT)-mcGAS[cat] structures at the active site (**B**) and the activation loop (**C**). PDB ID: 8shu (yellow) is from this work. The PDB ID for each structure is appended in parenthesis hereafter and Supplementary Tables 1–9 list all data collection and refinement statistics. **D** An overlay of dsDNA-free and dsDNA-bound WT-hcGAS[cat] structures. The "activation loop" in the new structure from this work (PDB ID: 8shz, yellow) is not modeled due to poor electron density. **E** dsDNA-free WT-hcGAS[cat] bound to cGAMP. The 2Fo-Fc map is contoured at 1.5 σ (same for all 2Fo-Fc maps shown hereafter). The Fo-Fc (omit) maps for all NTPs are shown in Fig. S8. **F** dsDNA-free WT-hcGAS[cat] bound to pppGpG. The triphosphate was modeled in the absence of electron densities for presentation. **G** dsDNA-free WT-mcGAS[cat] bound to ATP. The ribose was modeled without density for presentation.

formation. cGAS employs multiple proof-reading mechanisms including an intermediate-dependent nucleoside trap to prevent the cyclization of noncognate dinucleotides. Finally, we reveal how mouse cGAS achieves more stringent substrate selectivity and higher catalytic efficiency than the human enzyme. Together, we set forth a unifying catalytic mechanism of cGAS in which dsDNA-binding pre-organizes the active site into a rigid yet adaptable lock for two sets of keys (substrates and intermediate) necessary to specifically synthesize 2′–5/3′–5′-cGAMP.

## Results

### Activation of cGAS entails a disorder-to-order transition

Previous structural studies suggested that the active site of apo-cGAS is occluded from binding ATP/GTP (we denote a 1:1 mixture of NTPs and NTP/NTP hereafter). dsDNA binding reconfigures the "activation loop" and the three β-sheets that harbor catalytic acidic residues for coordinating NTPs and two $Mg^{2+}$ ions from the inactive to active conformation while allowing substrate binding[4,7]. We had intended to decipher why nucleic acids other than dsDNA bind cGAS but fail to activate the enzyme (e.g., refs. 11,12) by crystallizing the catalytic domain of mouse (m)cGAS[(cat)] with double-stranded (ds)RNA. However, mcGAS[cat] crystallized without the ligand, which instead provided us with an independent view of apo-mcGAS[cat] at 1.7 Å resolution. Here, the three β-sheets at the active site did not align particularly well with either dsDNA-free or dsDNA-bound mcGAS[cat] [4]. (Fig. 1B and

Supplementary Fig. 1A), while the activation loop aligned better with mcGAS[cat]•dsDNA (Ser[199]; Fig. 1B/C). Moreover, the active site appeared entirely solvent accessible, as the cavity volume was ~1000 Å$^3$ greater than that of mcGAS[cat]•dsDNA (Supplementary Fig. 1B). To further examine these discrepancies, we crystallized and solved the structure of the catalytic domain of human cGAS (hcGAS[cat]). We noted that, unlike the mouse enzyme, the three β-sheets do not move upon dsDNA binding, while the catalytic residues assume different rotamer conformations (Fig. 1D and Supplementary Fig. 1A)[13,14]. However, the activation loop in our structure was mostly disordered (thus not modeled in Fig. 1D and Supplementary Fig. 1A). Comparing the B-factors of hcGAS[cat] structures with or without bound dsDNA revealed that, while the C-lobe remains steady (likely via dimerization/crystal contact), dsDNA drastically stabilizes the N-lobe containing the activation loop and the catalytic residues (Supplementary Fig. 1C, D). Together, our observations suggest that, instead of being fixed into a specific auto-inhibited conformation[4,7], the active site of resting cGAS assumes an array of inactive states and its activation entails a dsDNA-dependent disorder-to-order transition.

### dsDNA modulates productive vs. nonproductive substrate binding

To understand why dsDNA-free cGAS is only basally active despite the open active site[15], we attempted to capture substrate-bound cGAS without dsDNA. After conducting co-crystallization and soaking trials,

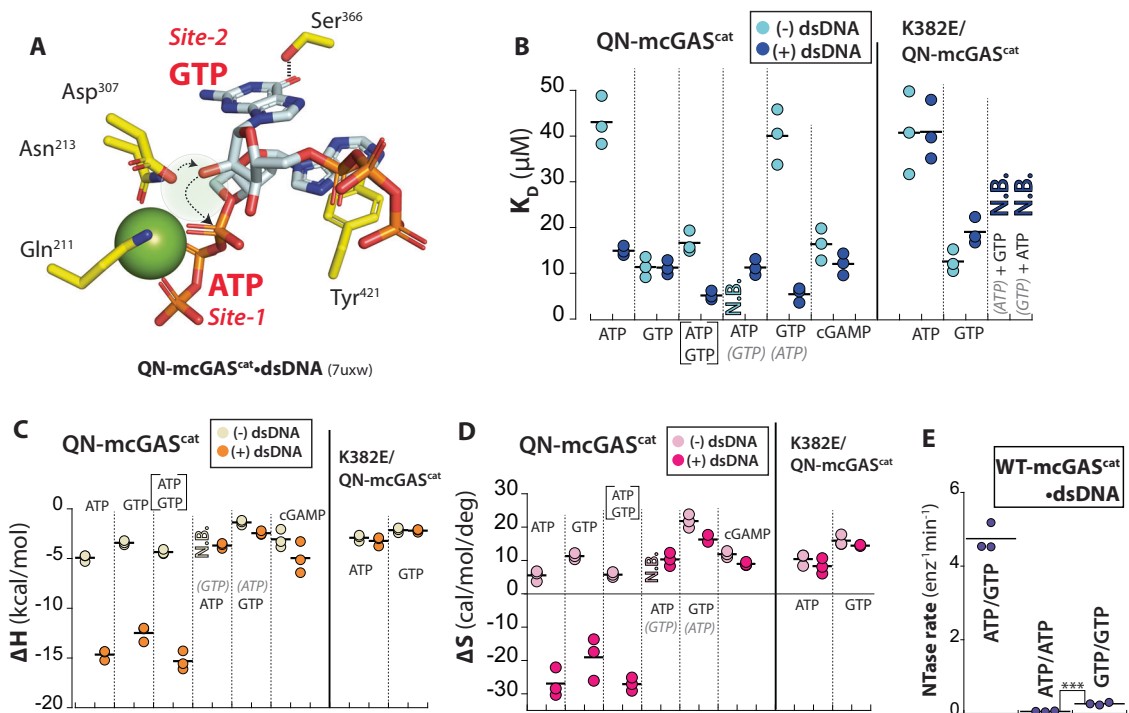

**Fig. 2 | dsDNA regulates productive vs. nonproductive substrate binding. A** The active site of E211Q/D213N (QN)-mcGAS$^{cat}$•dsDNA in complex with ATP and GTP. The position for the missing second Mg$^{2+}$ was shown as a shaded circle. Dotted arrows indicate the deprotonation and subsequent nucleophilic attack by the 2′-OH of GTP for the first linkage formation. The H-bond between Ser$^{366}$ and the carboxyl of GTP is also indicated. **B** The binding affinity ($K_D$) of mcGAS$^{cat}$ toward various NTPs with or without dsDNA (18-bp for all) was determined by ITC. The NTPs in gray parentheses are precomplexed with cGAS ± dsDNA. **C** The ΔH of mcGAS$^{cat}$ binding various NTPs with or without dsDNA. **D** The ΔS of mcGAS$^{cat}$ binding various NTPs with or without dsDNA ($T$ = 25 °C). **E** The catalytic activity of WT-mcGAS$^{cat}$•dsDNA (60-bp dsDNA) toward various NTPs. **p = 0.007. All p values hereafter are determined by Student's t test using Microsoft Excel (two-tailed with equal variances). *p < 0.05; **p < 0.01; ***p < 0.001.

we obtained the following structures with either human or mouse cGAS$^{cat}$. We co-crystallized hcGAS$^{cat}$ with Mg$^{2+}$ + ATP/GTP without dsDNA and found a new density corresponding to cGAMP at the active site; the catalytic residues remained in the inactive state (Fig. 1E; 5 mM Mg$^{2+}$ is present in all our soaking/biochemical experiments unless noted otherwise). Co-crystallizing hcGAS$^{cat}$ with GTP resulted in the pppGpG dinucleotide (GTP-GMP) bound in a conformation unamenable to further reaction: the densities for the triphosphate and Mg$^{2+}$ were missing and the catalytic residues remained in the inactive state (Fig. 1F). Also of note, the active site B-factors remain high despite bound NTPs in both structures (Supplementary Fig. 1D). We next solved the crystal structure of ATP-bound mcGAS$^{cat}$ without dsDNA via soaking. Here, one ATP was clearly bound at its designated binding pocket (Site-1), but the second NTP binding site (Site-2) remained empty (i.e., the GTP binding site; Fig. 1G). Strikingly, electron densities were missing for both the ribose of ATP and Mg$^{2+}$; the catalytic residues stayed in the inactive state and the triphosphate appeared suboptimal for coordinating Mg$^{2+}$ (Fig. 1G and Supplementary Fig. 1E). These results are consistent with the mechanism in which cGAS binds NTPs without dsDNA[15], but it is only basally active because it can rarely fix NTPs and metals into the catalytically competent conformation (i.e., nonproductive binding).

### dsDNA binding provides enthalpy for productive substrate binding

Our observation that dsDNA-free cGAS binds NTPs nonproductively prompted us to reinvestigate the role of dsDNA in catalysis. We largely employ mcGAS$^{cat}$ hereafter as its dsDNA-bound form was conducive to crystallization and binding assays using isothermal titration calorimetry (ITC). To generate a cGAS construct that binds NTPs but is

deficient in catalysis, we neutralized two acidic residues that coordinate Mg$^{2+}$•ATP[16] (E211Q-D213N in mcGAS, denoted as QN hereafter), which essentially abrogated the enzymatic activity (Supplementary Table 10). Soaking QN-mcGAS$^{cat}$•dsDNA crystals with ATP/GTP resulted in one Mg$^{2+}$ and each NTP bound in position for the first 2′−5′-linkage formation (Fig. 2A and Supplementary Fig. 2A; the loss of the second Mg$^{2+}$ was likely caused by the mutation, thus precluding catalysis). Soaking QN-mcGAS$^{cat}$•dsDNA crystals with GTP showed two GTPs in the catalytically competent state (Supplementary Fig. 2B). Moreover, soaking ATP to QN-mcGAS$^{cat}$ crystallized with and without dsDNA showed the productive (one Mg$^{2+}$) and nonproductive (no Mg$^{2+}$) binding states seen from WT ± dsDNA (two Mg$^{2+}$ and no Mg$^{2+}$), respectively (Fig. 1G and Supplementary Fig. 2C–E). These observations corroborate that the QN mutant is well-suited for delineating how dsDNA regulates substrate binding.

Consistent with our crystal structures (Fig. 1E−G), QN-mcGAS$^{cat}$ bound NTPs in μM affinities ($K_D$s) in the absence of dsDNA (Fig. 2B; values are listed in Supplementary Table 11 and representative ITC traces are shown in Supplementary Fig. 7). When QN-mcGAS$^{cat}$ was precomplexed with dsDNA, not only did the $K_D$ improve (Fig. 2B), but the overall thermodynamic profile also changed drastically (Fig. 2B–D). For example, when bound to dsDNA, the ΔH of ATP binding decreased by ~10 kcal/mol and the ΔS decreased by 30 cal/mol/deg. The favorable ΔΔH caused by dsDNA binding corroborates the formation of new interactions between the repositioned catalytic residues and Mg$^{2+}$•ATP, and the unfavorable ΔΔS supports the disorder-to-order transition observed in crystal structures (e.g., Supplementary Fig. 2C vs. 2D; e.g., ref. 17). Also of note, not only was cGAMP binding barely affected by dsDNA, but it also bound more weakly than ATP/GTP, suggesting that the product would not interfere with catalysis

(Fig. 2B–D). Combined with our crystal structures, these results suggest an activation mechanism in which dsDNA-free cGAS binds NTPs nonproductively; DNA binding affixes $Mg^{2+}$•NTPs into the catalytically competent conformation (favorable $\Delta\Delta H$ for bond formation and unfavorable $\Delta\Delta S$ for the decreased degree of freedom).

## Dimerization-coupled dsDNA-binding directs site-specific substrate binding

For the first linkage formation, Site-1 and Site-2 specifically bind ATP and GTP, respectively (Fig. 2A). Currently, mechanisms that direct the site-specific substrate binding remain unknown. Of note, ATP has always been found in Site-1 but not in Site-2 (Supplementary Fig. 2C–E and refs. 4,14), GTP appeared to be capable of binding at both Sites (Supplementary Fig. 2B), and each NTP was bound at its designated site upon soaking ATP/GTP (Fig. 2A and Supplementary Fig. 2A). Combined, these observations suggest that Site-1 intrinsically prefers ATP and Site-2 favors GTP while discriminating strongly against ATP. Indeed, WT-mcGAS$^{cat}$ showed the highest nucleotidyl transferase (NTase) activity toward ATP/GTP, yet it favored GTP alone (GTP/GTP) over ATP/ATP by 10-fold (Fig. 2E and Supplementary Table 10; i.e., the overall catalytic activity against any NTPs[15,18,19]).

dsDNA-dependent dimerization is tightly coupled to the activation of cGAS[6,15] (Fig. 1A). To determine how cGAS links dsDNA binding and dimerization to NTP binding, we titrated GTP to mcGAS$^{cat}$ pre-saturated with ATP and vice versa in the presence and absence of dsDNA in ITC. Sans dsDNA, GTP-bound QN-mcGAS$^{cat}$ failed to bind ATP, while pre-saturating the enzyme with ATP decreased GTP binding affinity by 4-fold (Fig. 2B–D). By contrast, each NTP bound to QN-mcGAS$^{cat}$•dsDNA pre-saturated with the other NTP even more tightly compared to the unliganded complex (Fig. 2B–D; the favorable $\Delta\Delta H$ and $\Delta\Delta S$ are consistent with NTP binding at correct positions and eventually displacing (partially) pre-bound NTPs), suggesting that dsDNA directs site-specific NTP binding. Additionally, without dsDNA, QN-mcGAS$^{cat}$ and a dimerization deficient mutant[6,15] (K382E/QN; see Supplementary Fig. 2F) bound ATP and GTP with similar affinities, indicating that dimerization per se is not required to bind NTPs (Fig. 2B–D). However, dsDNA failed to enhance the NTP binding of K382E/QN-cGAS$^{cat}$•dsDNA (Fig. 2B–D; e.g., the lack of favorable $\Delta\Delta H$ and now favorable $\Delta S$). Moreover, neither NTP bound to K382E/QN-cGAS$^{cat}$•dsDNA pre-saturated with the other (Fig. 2B). We thus surmised that dsDNA binding and dimerization are tightly coupled to direct site-specific (productive) ATP/GTP binding.

## Productive NTP binding underpins the distinction between ATP and deoxy-ATP at Site-1

When ATP is bound to mcGAS$^{cat}$•dsDNA at Site-1, its 2′-OH makes H-bonds with Glu$^{371}$ and Ser$^{368}$, while the 3′-OH interacts with Lys$^{424}$ and its own β-phosphate, apparently stabilizing the substrate (Fig. 3A and Supplementary Fig. 3A). Considering that the ribose of bound ATP was unresolved without dsDNA (Fig. 1G), we hypothesized that these interactions underpin productive substrate binding, which would also allow cGAS to discriminate against deoxy(d)-ATP. Indeed, mcGAS$^{cat}$•dsDNA displayed severely impaired NTase activities against 2′-dATP/GTP and 3′-dATP/GTP (Fig. 3B), revealing that the H-bonds from the 2′/ 3′-OH are important even for the first linkage formation. Although both dATPs bound to cGAS without dsDNA, the presence of the activator only marginally enhanced the $K_D$ and $\Delta H$ (Fig. 3B). Moreover, compared to when it was pre-bound with ATP, QN-mcGAS$^{cat}$•dsDNA pre-saturated with either dATP bound GTP at least 5-fold more weakly (Fig. 3B vs. Fig. 2B). To determine the mechanism, we soaked 2′- or 3′-dATP to WT-mcGAS$^{cat}$•dsDNA crystals. Here, the catalytic residues retained the active conformation; however, the ribose of bound 2′-dATP was flipped out of the active site, apparently resulting in distorted α-

phosphate and the loss of the second $Mg^{2+}$ (Fig. 3C). Furthermore, similar to nonproductively bound ATP in dsDNA-free cGAS (Fig. 1G), the densities for the ribose of 3′-dATP and the second $Mg^{2+}$ were missing (Supplementary Fig. 3B). These results demonstrate that cGAS utilizes intricate protein•NTP and intra-NTP interactions for productive substrate binding, concomitantly permitting the distinction between ATP and dATPs.

## Molecular basis for distinguishing ATP from GTP at Site-1

Two GTPs bind essentially in the same manner as ATP/GTP in QN-mcGAS$^{cat}$•dsDNA (Supplementary Fig. 2A, B). Nevertheless, although Tyr$^{421}$ stacks on both ATP and GTP, unlike ATP, the -NH$_2$ of guanine was in proximity to His$^{467}$ near Site-1 (Fig. 3D; ≤3 Å). We reasoned that, in our buffer conditions (pH 7.4 for reaction (i.e., physiological) and 6.5 for crystallization), the interaction between the -NH$_2$ (H-bond donor) of guanine and the partially protonated imidazole ring of His$^{467}$ could be unfavorable due to a steric clash. Moreover, in the ATP/GTP-bound structure, the -NH$_2$ of ATP at Site-1 appeared to donate a H-bond to the α-phosphate oxygen of GTP at Site-2; however, such an inter-NTP interaction beyond base-stacking is unlikely when GTP binds at Site-1 (Fig. 3D; the carboxyl oxygen of GTP is a H-bond acceptor). To test functional implications, we soaked QN-mcGAS$^{cat}$•dsDNA with inosine triphosphate (ITP)/GTP or ATP/ITP. In the ITP/GTP-soaked structure, judged by the electron density for the -NH$_2$ of GTP, it appeared that ITP and GTP bind at Site-1 and Site-2, respectively (Fig. 3E). Moreover, the ATP/ITP-soaked structure suggested that ATP and ITP are bound at Site-1 and Site-2, respectively (Fig. 3F); the carboxyl oxygen of ITP appeared to accept a H-bond from Ser$^{366}$ as seen from GTP, and ATP fails to bind stably at Site-2 in all known structures (see also below for the pppGpA-bound structure). These results support the idea that the -NH$_2$ of GTP is disfavored at Site-1 while the -NH$_2$ of ATP at Site-1 provides a favorable interaction for site-specific ATP/GTP binding. Consistent with the idea that GTP is disfavored over ITP at Site-1, mcGAS$^{cat}$ was more active toward ITP/ITP vs. GTP/GTP (Fig. 3G). Moreover, supporting the idea that His$^{467}$ interferes with GTP binding at Site-1, although H467A-mcGAS$^{cat}$ was less active toward ATP/GTP than WT, it showed a 2-fold higher activity toward GTP/GTP (Fig. 3H). Resolving reaction products via HPLC revealed that, unlike WT, H467A-mcGAS$^{cat}$ produced off-pathway pppGpG even with ATP/GTP (Fig. 3I); the H467A mutant was indeed more active toward GTP/GTP (Fig. 3J, K). Additionally, supporting that the -NH$_2$ of ATP contributes positively to the recognition of the ATP/GTP pair by donating the H-bond to the α-phosphate of GTP, mcGAS$^{cat}$ showed a significantly higher NTase activity toward ATP/GTP vs. purine triphosphate (PuTP)/GTP or ATP/PuTP (Fig. 3G; PuTP lacks the NH$_2$ of ATP).

## The GTP binding at Site-2 dictates the signature 2′-5′-linkage specificity

It is thought that Thr$^{197}$ at Site-2 recognizes the -NH$_2$ of GTP, thereby prescribing its specificity[4] (Fig. 4A). However, not only did mcGAS$^{cat}$ show a robust NTase activity toward ATP/ITP, but ITP/ITP was also more favored over GTP/GTP (Fig. 3G), indicating that the interaction between the -NH$_2$ of GTP and Thr$^{197}$ is not critical. We noted that Arg$^{364}$ appears to stabilize GTP at Site-2 by donating H-bonds at the Hoogsteen edge (Fig. 4A). However, the guanidium of Arg$^{364}$ would repel the positive dipole of adenine -NH$_2$, thereby suppressing ATP binding at Site-2. Supporting this notion, ATP/PuTP showed 3-fold higher NTase activity vs. ATP/ATP (Fig. 3G). Moreover, compared to WT, R364A-mcGAS$^{cat}$•dsDNA showed 33-fold and 10-fold lower NTase activities against ATP/GTP and GTP/GTP, respectively; however, it was 4-fold more active toward ATP/ATP (Fig. 4B). Additionally, unlike WT-mcGAS$^{cat}$ which generated 2′-5′-linked pppApA, R364A-mcGAS$^{cat}$ predominantly produced 3′-5′-linked pppApA (Fig. 4C; S1 nuclease resistant vs. susceptible).

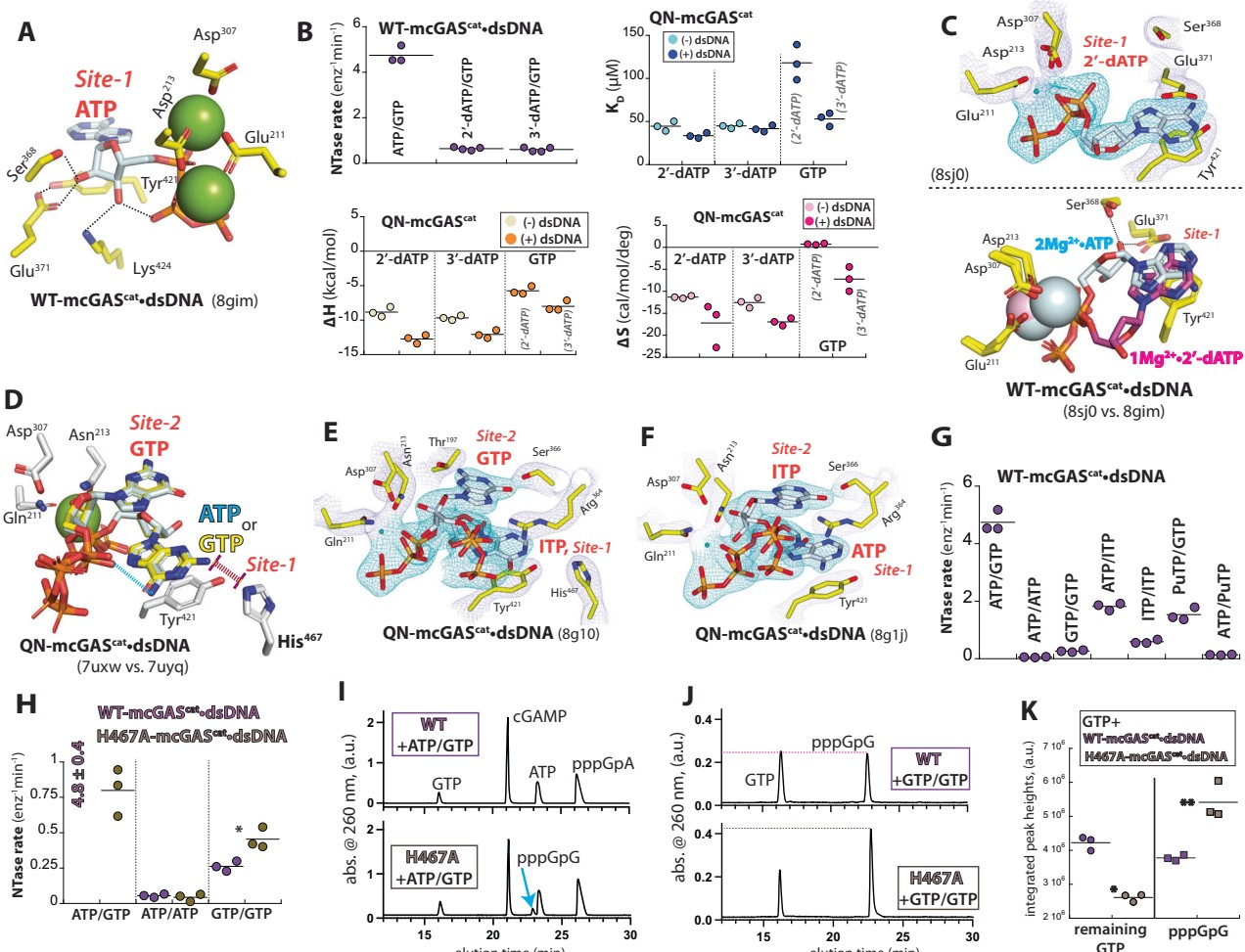

**Fig. 3 | Molecular bases for ATP recognition at Site-1. A** An "interior view" of the active site of ATP-bound WT-mcGAS$^{cat}$•dsDNA showing protein•NTP and intra-NTP interactions. **B** Catalytic activity, $K_D$, and $\Delta H/\Delta S$ values for mcGAS$^{cat}$ ± dsDNA when deoxy-ATPs were used as a substrate. dATPs in gray parentheses are precomplexed with mcGAS$^{cat}$•dsDNA. Colored symbols indicate experiments with designated conditions hereafter (e.g. ± dsDNA, or WT vs. a specific mutant). **C** 2'-dATP bound to WT-mcGAS$^{cat}$•dsDNA. Bottom: an overlay of ATP- vs. 2'-dATP-bound structures. **D** An overlay of ATP/GTP-bound and GTP/GTP-bound QN-mcGAS$^{cat}$•dsDNA. The H-bond donation from the -NH$_2$ of adenine plus the steric clash between the -NH$_2$ of guanine and His$^{467}$ are indicated. **E, F** QN-mcGAS$^{cat}$•dsDNA bound to ITP/GTP (**E**) and ATP/ITP (**F**). **G** The catalytic activity of WT-mcGAS$^{cat}$•dsDNA toward various NTPs. **H** The catalytic activity of WT- and H467A-mcGAS$^{cat}$•dsDNA toward various NTPs. *$p$ = 0.017. **I, J** WT- and H467A-mcGAS$^{cat}$•dsDNA reaction products resulting from various NTPs (1 hr reaction time) were resolved via high-pressure liquid chromatography (HPLC, 37 min gradient); a.u.: arbitrary units. **K** The comparison of GTP consumption and pppGpG generation between WT- and H467A-mcGAS$^{cat}$•dsDNA. Shown p-values were determined using a two-tailed t-test with equal variances, hereafter. *$p$ = 0.025; **$p$ = 0.001.

Although the second Mg$^{2+}$ is absent in our ATP/GTP-bound QN-mcGAS$^{cat}$•dsDNA structure, not only is the 2'-OH of GTP already inline to attack the α-phosphate of ATP at Site-1, but it also makes a H-bond with Asp$^{307}$, the residue that deprotonates the nucleophile (Fig. 4A). These interactions then place the 3'-OH of GTP too far to interact with Asp$^{307}$ (4 Å), barring it from participating in the first linkage formation. Indeed, mcGAS$^{cat}$ was essentially inactive toward ATP/2'-dGTP in our NTase assay (Supplementary Table 10). Moreover, analogous to 2'/3'-dATP, dsDNA failed to enhance the affinity or ΔH of 2'-dGTP binding; QN-mcGAS$^{cat}$•dsDNA pre-saturated with ATP also failed to enhance 2'-dGTP binding (Fig. 4D vs. Fig. 2B). These results suggest that the H-bond between Asp$^{307}$ and the 2'-OH of GTP is integral to dsDNA-dependent productive substrate binding. To further delineate the mechanism, we soaked WT-mcGAS$^{cat}$•dsDNA crystals with ATP/2'-dGTP. Here, although 2'-dGTP at Site-2 was bound in the same manner as GTP (and 1Mg$^{2+}$•ATP at Site 1; Fig. 4E), the densities for the second Mg$^{2+}$ and the bottom half of the deoxyribose were missing (Fig. 4E). These results consistently suggest that pre-establishing the H-bond between Asp$^{307}$

and the 2'-OH of GTP is essential for productive substrate binding, prescribing the signature 2'-5' linkage specificity, and capturing/retaining the second divalent metal.

## Mn$^{2+}$ is preferentially incorporated as the catalytic metal

Although cGAS employs two divalent metals in catalysis[4], their roles remain poorly defined. Our structures consistently suggest that only one Mg$^{2+}$ is necessary for productive substrate binding (e.g., Fig. 4A), while the second Mg$^{2+}$ drives chemistry (i.e., neutralizing negative charges upon deprotonating 2'- or 3'-OH by Asp$^{307}$). Of note, cytosolic Mn$^{2+}$ released from damaged organelles accentuates the dsDNA sensing activity of cGAS in cells and in vivo[20]. Furthermore, it was reported that Mn$^{2+}$ activates cGAS in the absence of dsDNA without involving the canonical catalytic residues by invoking an inverted intermediate, albeit necessary [Mn$^{2+}$] is unphysiologically high[21] (>1 mM vs. ≤ 50 μM in vivo). Nevertheless, although we also found that high [Mn$^{2+}$] can activate cGAS without dsDNA[18], we showed that excess Mg$^{2+}$ and dsDNA binding synergistically allow cGAS to incorporate physiological concentrations of Mn$^{2+}$ to boost its catalytic activity[18]. To elucidate the

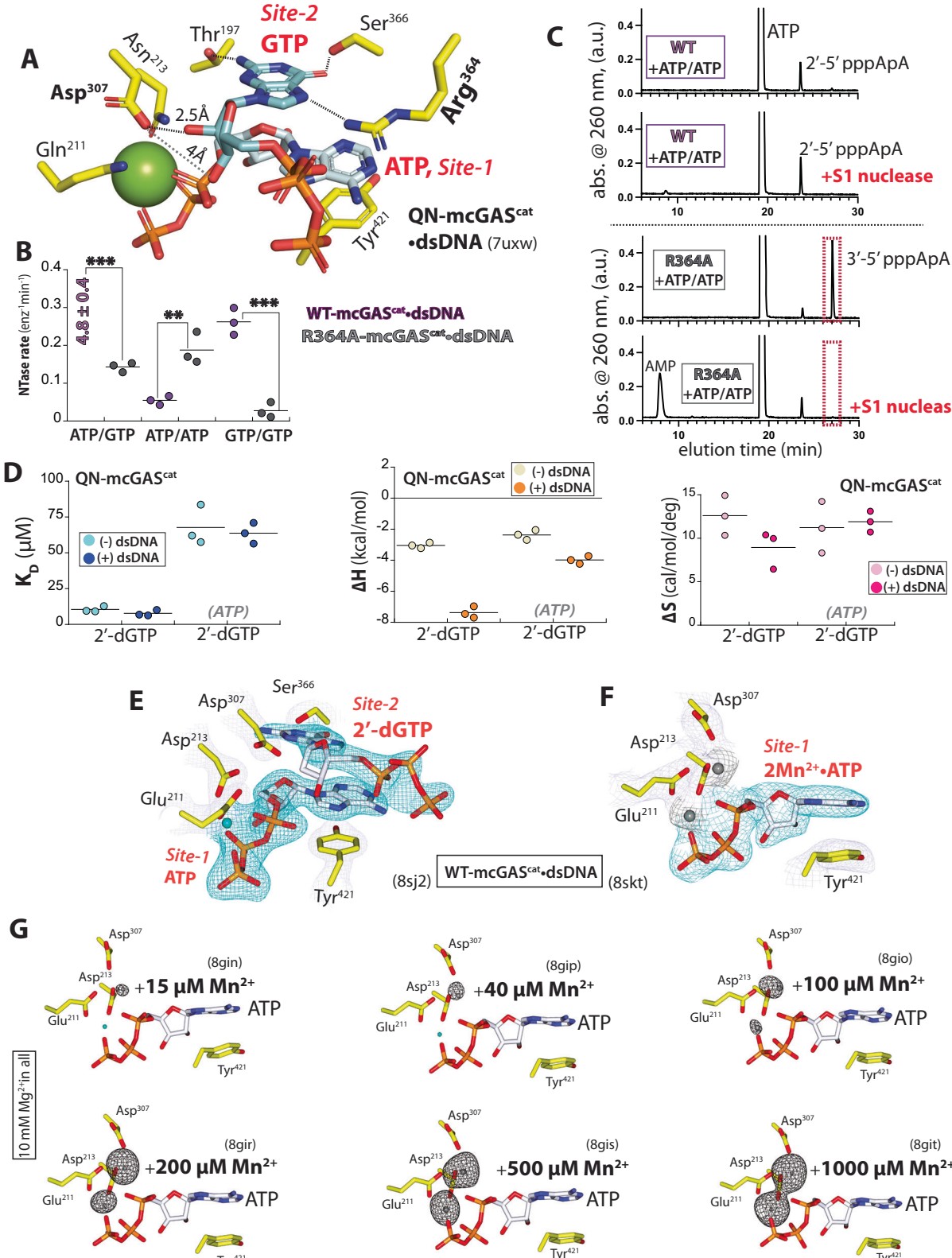

**Fig. 4 | Mechanisms that define the GTP specificity at Site-2 and Mn²⁺ utilization. A** ATP/GTP-bound QN-mcGAS$^{cat}$•dsDNA. **B** The catalytic activity of WT- and R364A-mcGAS$^{cat}$•dsDNA against various NTPs. $p$ values from left to right: ***$p = 0.00003$; **$p = 0.006$; ***$p = 0.0006$. **C** WT- and R364A-mcGAS$^{cat}$•dsDNA reaction products resulting from ATP/ATP were resolved via HPLC (2 h reaction, 27 min gradient). The S1 nuclease specifically degrades 3′−5′-linked oligonucleotides. **D** $K_D$ and ΔH/ΔS values of mcGAS$^{cat}$ toward 2′-dGTP with or without dsDNA.

ATP in the gray parenthesis is precomplexed with mcGAS$^{cat}$ ± dsDNA. **E** 1Mg²⁺•ATP/2′-dGTP-bound WT-mcGAS$^{cat}$•dsDNA. **F** ATP-bound WT-mcGAS$^{cat}$•dsDNA in the presence of 5 mM Mn²⁺ without any Mg²⁺. **G** The presence of Mn²⁺ at each indicated concentration for the ATP-bound WT-mcGAS$^{cat}$•dsDNA crystal was monitored by tracking its anomalous signal (contoured at 5σ; the average resolution of maps shown: 2.63 ± 0.11 Å; see also Supplementary Tables 8 and 9).

mechanism by which dsDNA-bound cGAS incorporates $Mg^{2+}$ and $Mn^{2+}$ in catalysis, we first soaked WT-mcGAS$^{cat}$•dsDNA crystals with $Mn^{2+}$ + ATP and found 2$Mn^{2+}$•ATP as seen from the 2$Mg^{2+}$•ATP-bound structure (Fig. 4F and Supplementary Fig. 2E). Next, to determine whether $Mn^{2+}$ is preferred at either metal binding site, we soaked WT-mcGAS$^{cat}$•dsDNA crystals with ATP pre-complexed with increasing concentrations of $Mn^{2+}$ (0.015-1 mM) in the presence of excess $Mg^{2+}$ (10 mM); the presence of $Mn^{2+}$ was tracked via its anomalous signal at 1.8961 Å (6539 eV) during data collection. Here, we found that cGAS specifically favors $Mn^{2+}$ over $Mg^{2+}$ by ≥100-fold at the catalytic metal binding site (Fig. 4G); the binding here also cooperatively increased the affinity for $Mn^{2+}$ at the first site (Fig. 4G and Supplementary Fig. 3C). Together, we concluded that $Mn^{2+}$ accelerates the signaling activity of cGAS•dsDNA by preferentially incorporated as the catalytic metal over $Mg^{2+}$ without requiring any other reaction pathways.

### Mechanisms that govern the second 3′-5′ linkage specificity and product fidelity

cGAS can generate linear dinucleotides from various NTPs, but it remains unknown why only ATP/GTP can be readily cyclized[4,5,18]. Indeed, cGAS failed to produce cyclic dinucleotides from GTP/GTP, ATP/ATP, ATP/ITP, and ITP/GTP (Figs. 3J and 4C and Supplementary Fig. 4). To decipher the cyclization mechanism, we sought to capture the cognate pppGpA intermediate in crystallo. We noted that $Mn^{2+}$ can activate QN-mcGAS$^{cat}$•dsDNA to some extent (the catalytic Asp$^{307}$ that deprotonates the OH for nucleophilic attack is still intact). Soaking these crystals with ATP/GTP precomplexed with 5 mM $Mg^{2+}$/ 1 mM $Mn^{2+}$ resulted in 1$Mn^{2+}$•pppGpA trapped in the active site (Fig. 5A and Supplementary Fig. 5A; the second metal was missing likely due to mutation). Strikingly, compared to the guanine of GTP at Site-2, the adenine of pppGpA was rotated ~30°, exposing the N1 imine as the H-bond acceptor from Ser$^{366}$ instead of the carboxyl of guanine (Fig. 5A, B; the N1 amide in guanosine is a H-bond donor). Consequently, the 3′-OH of adenosine was aligned to be deprotonated by Asp$^{307}$ for the subsequent nucleophilic attack, while the 2′-OH was completely occluded (Fig. 5A, B). Moreover, Arg$^{364}$ was pushed away by the repulsive positive dipole of adenine (Fig. 5A, B). Similar to the β-phosphate•3′-OH interaction seen from productive ATP binding, the -NH₂ of guanine was then no longer close to His$^{467}$ but appeared to donate an intramolecular H-bond to the α-phosphate to stabilize the intermediate for cyclization (Fig. 5A).

To test whether such a precise NTP-specific coordination is critical for cyclization, we soaked $Mg^{2+}$+ITP/GTP, ATP/ITP, or GTP alone into WT-mcGAS$^{cat}$•dsDNA, all of which produced respective linear dinucleotides bound at the active site. First, pppGpI (Fig. 5C) was bound analogous to the previously reported pppGpG[4] (Supplementary Fig. 5B, C). Here, the catalytic $Mg^{2+}$ was absent, and unlike pppGpA where the 2′-OH was occluded, both 2′-OH and 3′-OH were near Asp$^{307}$ (Fig. 5C bottom and Supplementary Fig. 5C), which likely have sterically hindered the second metal binding (also possibly obscuring the deprotonation target). Next, we identified pppIpA based on how the nucleoside at Site-2 resembles the adenine of pppGpA (Fig. 5D and Supplementary Fig. 5D); Arg$^{364}$ was also pushed away as seen from pppGpA and the 3′-OH was in position for catalysis. Nevertheless, the second $Mg^{2+}$ was again missing, and the α-phosphate of pppIpA appeared misaligned compared to pppGpA (Fig. 5D, bottom). These observations suggest that the H-bond between the α-phosphate and the -NH₂ of guanine is important for positioning the metal•intermediate complex for cyclization. Even more strikingly, when we soaked GTP alone, we found a different conformation of pppGpG in which the guanosine was no longer bound at Site-2 but trapped via intra-NTP H-bonds despite both $Mg^{2+}$ ions being present for catalysis (Fig. 5E and Supplementary Fig. 5E). Soaking $Mn^{2+}$•GTP yielded the same "flipped-out" pppGpG seen from soaking $Mg^{2+}$•GTP (Supplementary Fig. 5F), indicating that this apparent proof-reading

mechanism is not metal-dependent. Together, we concluded that cGAS precisely positions the cognate pppGpA intermediate not only for the 3′-5′ linkage specificity, but also for the likelihood of cyclization.

### Human and mouse cGAS enzymes display different active site reactivity and fidelity

Considering the primary sequence similarity at the active site (only three residue differences between mouse and human: Ile$^{309}$/Thr$^{321}$, Cys$^{419}$/Ser$^{434}$, and His$^{467}$/Asn$^{482}$), we propose that our findings are broadly relevant across vertebrates. Of note, it was recently reported that recombinant mcGAS$^{cat}$ is more active than hcGAS$^{cat}$ because the former relies less on dsDNA length for binding/activation[14]. However, others have reported that mcGAS and hcGAS display similar dsDNA length-dependence[22]. Indeed, we found that recombinant mcGAS still binds dsDNA in a length-dependent manner (Fig. 6A); however, the mouse enzyme was at least 5-fold more active than hcGAS (Fig. 6B). Interestingly, compared to hcGAS$^{cat}$, mcGAS$^{cat}$ was more active toward GTP/GTP, but less active toward ATP/ATP (Fig. 6C); the pppApA linkage was predominantly 3′-5′ for hcGAS$^{cat}$ unlike mcGAS$^{cat}$ (Supplementary Fig. 6). These results suggested the two cGAS enzymes have different active site reactivity and promiscuity. We then noted that Ile$^{309}$ near GTP at Site-2 is Thr$^{321}$ in hcGAS (Fig. 6D), the former of which could stack better against the base than the latter. Indeed, T321I-hcGAS$^{cat}$ showed markedly higher NTase activities against ATP/GTP and GTP than WT, while that for ATP was reduced (Fig. 6C). Moreover, pppApA resulting from T321I-hcGAS$^{cat}$ was 2′-5′ linked (mcGAS-like), while I309T-mcGAS$^{cat}$ produced an increased amount of the 3′-5′-linked dinucleotide (hcGAS-like; Supplementary Fig. 6). These observations indicate that Ile$^{309}$ of mcGAS suppresses ATP binding while stabilizing GTP at Site-2.

We noted that T321I-hcGAS$^{cat}$ is defective in cyclization although it efficiently generates the pppGpA intermediate (Fig. 6E). Further examining pppGpA-bound mcGAS$^{cat}$ indicated that Cys$^{419}$, which forms a H-bond with the ribose moiety of guanosine, is Ser in human (Fig. 6D). T321I/S431C-hcGAS$^{cat}$ was not only even more active (Fig. 6C), but also readily cyclized pppGpA into cGAMP (Fig. 6E). Finally, when transfected into HEK293T cells, mcGAS$^{FL}$ and hcGAS$^{FL}$ showed comparable human STING-dependent IFN-Luc (luciferase) activities (Fig. 6F), indicating that in cellulo signaling outputs of cGAS from different species are likely similar despite varying catalytic potentials at the active site. Nevertheless, consistent with our biochemical assays, the signaling activity of T321I- hcGAS$^{FL}$ was compromised, but T321I/S431C-hcGAS$^{FL}$ displayed a significantly higher IFN-Luc activity (Fig. 6F). Together, we concluded that human and mouse cGAS enzymes have different active site reactivity and promiscuity. Moreover, we identify additional key interactions that regulate the site-specific substrate binding (Ile$^{309}$ for GTP binding at Site-2 and ensuring the 2′-5′ linkage) and the cyclization efficiency (Cys$^{419}$ for stabilizing pppGpA).

## Discussion

Not only is cGAS essential to the innate immune system[1–3], but cGAS-like receptors are also central to the host defense across a wide range of organisms[9]. We present here a unifying catalytic mechanism of mammalian cGAS in which the dsDNA-dependent disorder-to-order transition generates a rigid yet adaptable lock to specifically recognize its cognate substrates and intermediate (Fig. 7).

### The role of dsDNA and dimerization in cGAS activation

Resting cGAS would assume an array of inactive conformations without being fixed into a specific autoinhibited state (Figs. 1 and 7). NTPs can bind cGAS without dsDNA; however, the enzyme remains only basally active as it can rarely achieve the site-specific metal•NTP binding necessary for catalysis (Figs. 1 and 2). Dimerization-coupled

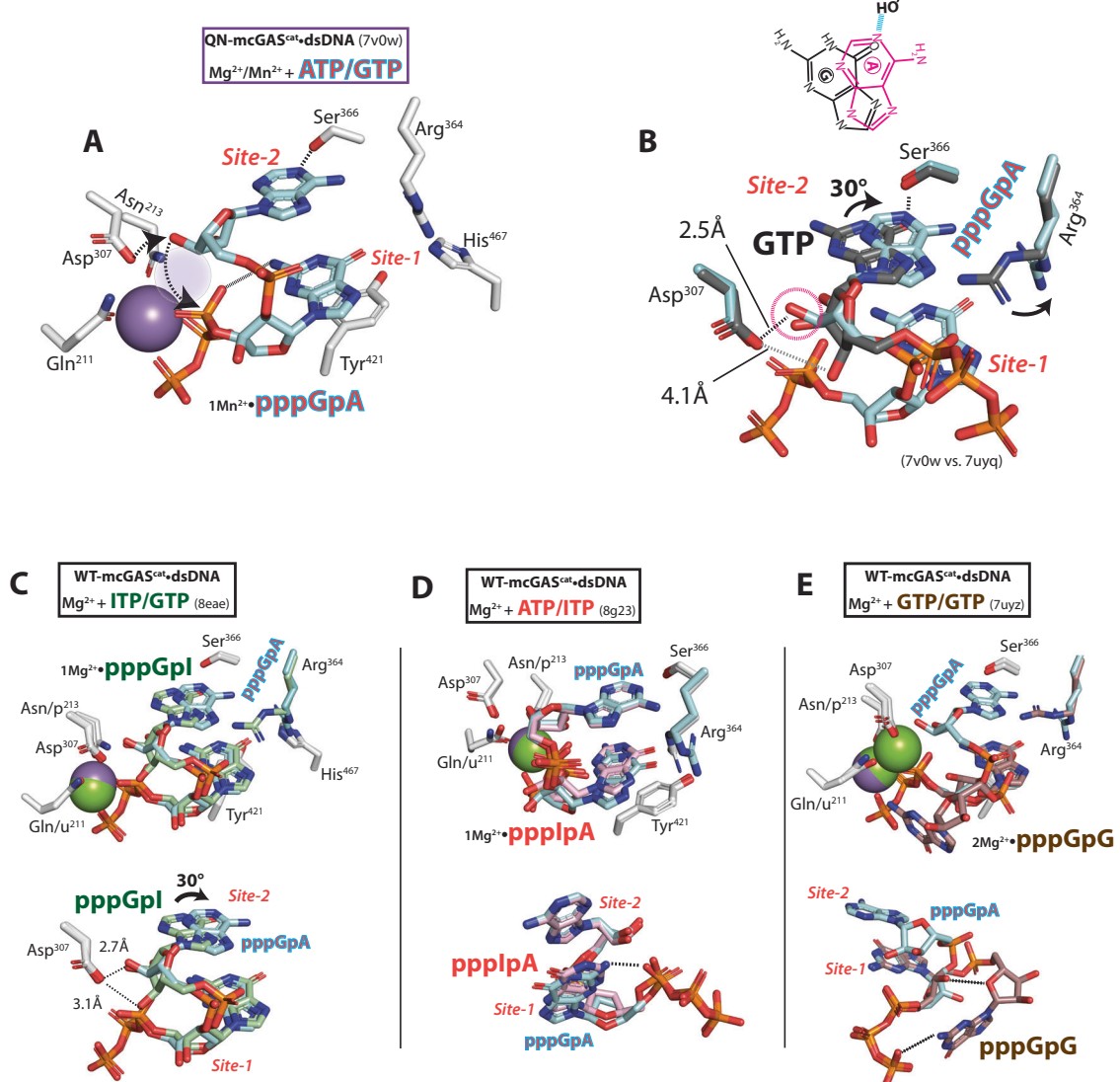

**Fig. 5 | Molecular determinants that drive the selective cyclization of pppGpA.**
**A** 1Mn²⁺•pppGpA-bound QN-mcGAS^cat•dsDNA. The position for the missing second metal is shown as a shaded circle. Dotted arrows indicate the deprotonation of 3′-OH of the intermediate and the subsequent nucleophilic attack for cyclization. The intramolecular H-bond within GTP and the H-bond between adenine and Ser³⁶⁶ are also indicated. **B** Comparison between GTP-bound and pppGpA-bound

mcGAS^cat•dsDNA at Site-2. The -30° rotation between guanine and adenine at Site-2 and the new position of Arg³⁶⁴ are indicated. The chemical structures of adenine and guanine at the respective bound positions are also shown (top).
**C–E** Comparing pppGpA-bound QN-mcGAS^cat•dsDNA vs. WT-mcGAS^cat•dsDNA bound to pppGpI (**C**), pppIpA (**D**), and pppGpG (**E**). The bottom figures in **C–E** show an alternate/simplified view.

dsDNA binding then establishes a web-like network of highly sensitive protein•NTP, intra-NTP, and inter-NTP interactions leading to productive nucleotide transactions (Figs. 2 and 4).

**Mechanisms that underpin NTP and linkage specificities**
We propose that the productive 1Mg²⁺•triphosphate binding at Site-1 provides the most significant interaction energy (ΔH; Fig. 2). The intramolecular H-bond between the β-phosphate and the 3′-OH of ATP stabilizes its α-phosphate electrophile and the second divalent metal for catalysis, while precluding any reaction with dATP (Fig. 3). Moreover, His⁴⁶⁷ sterically suppresses GTP binding at Site-1, while ATP binding here allows its differentially positioned -NH₂ (positive dipole) to interact favorably with the triphosphates of GTP at Site-2 (Fig. 3). Considering that ATP outnumbers GTP by ~6-fold in vivo[23], the additional 18-fold preference for ATP created by these interactions (3-fold from ATP/GTP vs. PuTP/GTP for WT and ~6-fold from WT vs. H476A against ATP/GTP; Fig. 3G) would synergistically

increase the likelihood of ATP binding at Site-1 by ~110-fold over GTP. At Site-2, Ser³⁶⁶ acts as the most energetically important H-bond donor to GTP, and Ile³⁰⁹ and Arg³⁶⁴ fix GTP in place for the 2′-5′ bond formation while discriminating against ATP (Figs. 4, 6, and 7). Also of note, pre-establishing the H-bond between the 2′-OH of GTP and Asp³⁰⁷ is critical not only for aligning the obligate nucleophile for the first linkage formation, but also for retaining the catalytic divalent metal (Figs. 4 and 7).

For cyclization, pppGpA would also be first recognized by the favorable 1Mg²⁺•triphosphate•enzyme interactions at Site-1. This anchoring interaction then allows the adenine of the intermediate to accept a H-bond from Ser³⁶⁶ at Site-2 while overcoming the Ile³⁰⁹/Arg³⁶⁴ blockade (Figs. 5–7). This binding mechanism then precisely positions the 3′-OH of adenosine for cyclization while excluding its 2′-OH (Fig. 5). Additionally, similar to the 3′-OH and β-phosphate of ATP, the -NH₂ of guanine (pppGpA) is crucial for correctly positioning the target α-phosphate to retain the catalytic metal for cyclization; Cys⁴¹⁹ also helps

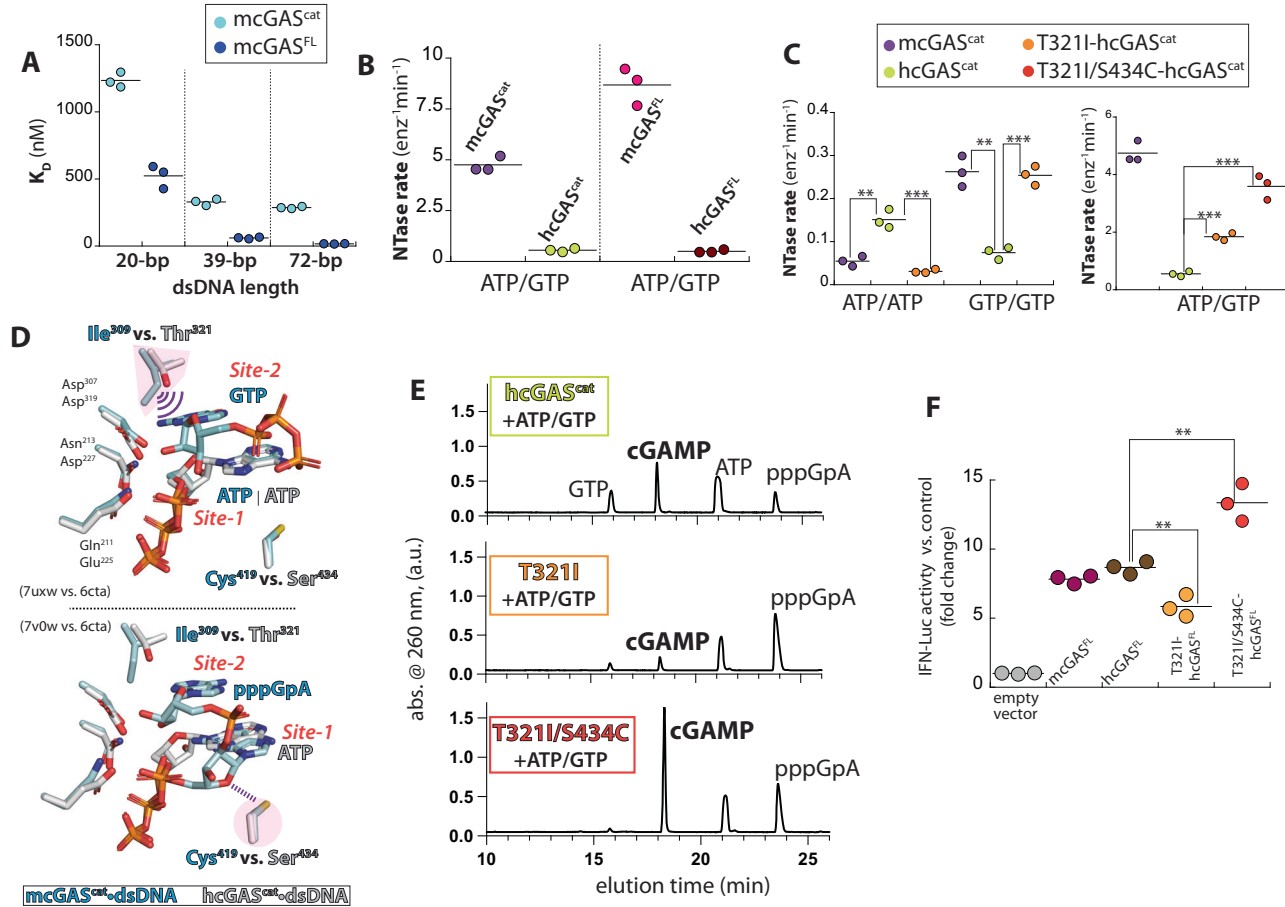

**Fig. 6 | Mechanisms that prescribe the intrinsically higher catalytic activity of mcGAS vs. hcGAS. A** The binding affinities of mcGAS[cat] and full-length mcGAS[(FL)] toward various lengths of FAM-labeled dsDNA were determined by tracking the changes in fluorescence anisotropy (FA). **B** The catalytic activity of mcGAS and hcGAS. **C** Catalytic activity of mcGAS[cat] vs. WT- and "mouseneized"-hcGAS[cat] toward various NTPs. $p$ values from left to right: **$p = 0.0024$; ***$p = 0.0007$; **$p = 0.001$; ***$p = 0.0003$; ***$p = 0.0001$; ***$p = 0.0003$. **D** An overlay of ATP/GTP-bound or pppGpA-bound mcGAS[cat]•dsDNA (7uxw and 7v0w) vs. ATP-bound hcGAS[cat]•dsDNA (6cta). **E** WT- and "mousenized"-hcGAS[cat]•dsDNA reaction products resulting from ATP/GTP were resolved by HPLC (1 hr reaction time, 27 min gradient). **F** IFN-luc reporter activities from HEK293T cells upon co-transfecting STING and indicated cGAS variants. $p$ values from left to right: **$p = 0.006$; **$p = 0.005$.

stabilizing the intermediate (Figs. 5–6). Importantly, our results consistently indicate that any deviations from such a precise coordination would prevent cyclization (Figs. 5 and 7): any misalignments of the 3′-OH nucleophile (pppGpI) or the α-phosphate electrophile (pppIpA) would disrupt catalytic metal binding, and the enzyme-mediated intra-dinucleotide interactions would trap pppGpG in a nonproductive state.

### Metal utilization

We show that capturing/retaining the second divalent metal is crucial for distinguishing NTPs from d-NTPs, defining the signature 2′-5′-linkage, and selectively cyclizing the cognate intermediate (Figs. 3–5 and 7). Moreover, $Mn^{2+}$ is specifically preferred as the second catalytic metal over $Mg^{2+}$ (Fig. 4F-G), explaining how the emergence of cytosolic $Mn^{2+}$ from damaged organelles can potentiate the signaling activity of cGAS[20]. Importantly, when cGAS is bound to dsDNA, $Mn^{2+}$-mediated catalysis does not invoke alternate mechanisms (Fig. 4F-G). We reason that $Mn^{2+}$ being a transitional metal with more flexible polarizable options than alkaline $Mg^{2+}$ likely allows the former to bind more tightly as the catalytic metal.

### Of human and of mice

In recent years, species-dependent specificities of innate immune pathways have garnered much attention. For instance, small molecule modulators arising from targeting mcGAS and mouse STING

have failed to be effective toward the human counterparts[24–27]. We show that, compared to hcGAS, not only does mcGAS enforce a stricter GTP specificity at Site-2 (Ile[309] vs. Thr[321]; Fig. 6), but it also cyclizes pppGpA more efficiently (Cys[419] vs. Ser[434]; Fig. 6). Nonetheless, our cellular assays suggest that the species-dependent active site reactivity per se does not dictate the overall signaling output of cGAS (Fig. 6F). It is still possible that there might be small but functional differences that our transfection system failed to capture. Nonetheless, we speculate that other factors such as higher dimerization propensity[15] and the ability to form phase-separated condensates[11] compensate hcGAS for its lower catalytic efficiency in vivo.

Future studies focusing on detailed kinetic analyses coupled with molecular dynamics simulations will provide a deeper understanding of cGAS catalytic mechanisms, which include exactly what the rate-limiting step is and how the intermediate is released and recaptured for cyclization.

## Methods

### Protein expression and purification

Human cGAS (full-length (hcGAS[FL]) and the catalytic domain (hcGAS[cat], residue 157-522)) and mouse cGAS (mcGAS[FL] and mcGAS[cat] (residue 147-507)) were cloned into the pET28b vector (Novagen) with an N-terminal 6×His-MBP-tag containing a TEV protease cleavage site. Plasmids encoding wild-type and mutant cGAS constructs

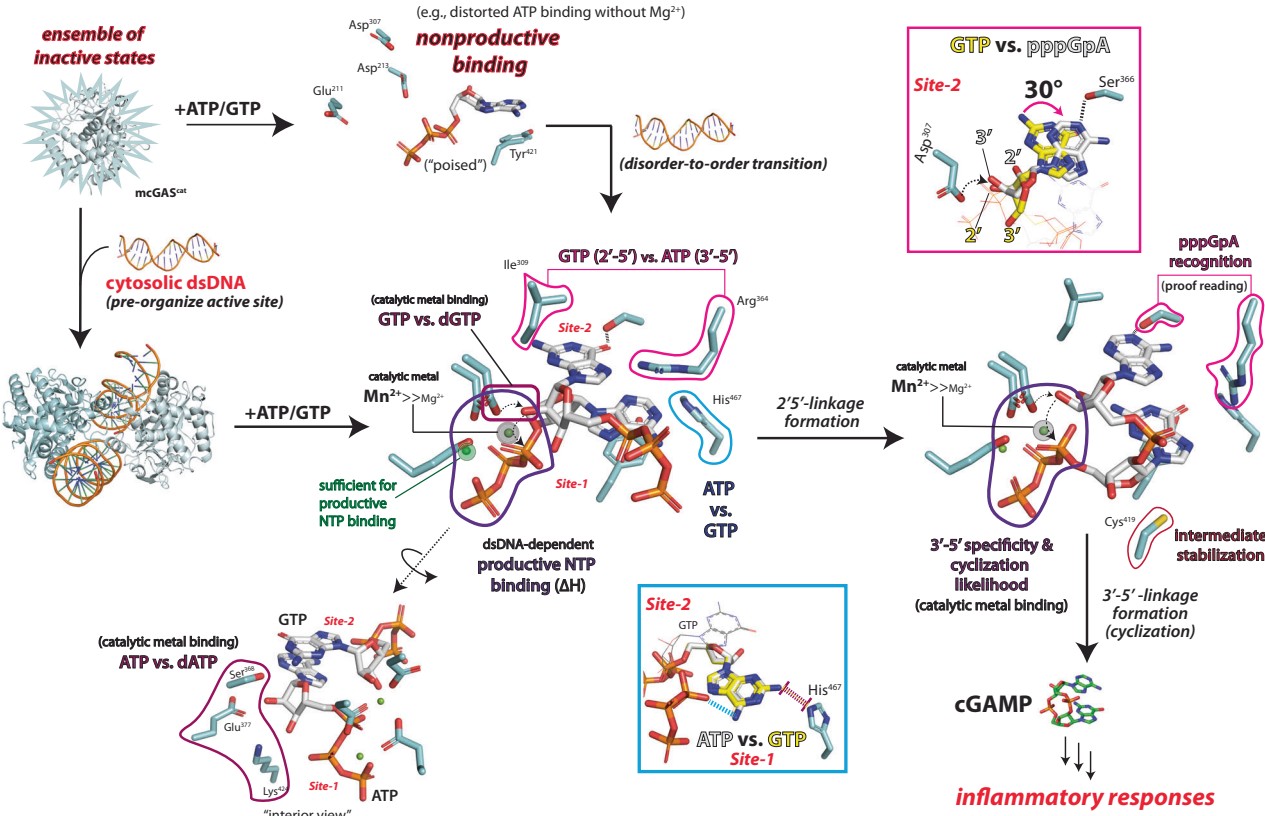

**Fig. 7 | The catalytic mechanism of cGAS.** Resting cGAS assumes an ensemble of inactive conformations and binds NTPs nonproductively. Dimerization-coupled dsDNA-binding affixes cGAS into the active form, allowing site-specific substrate/metal binding. Only one $Mg^{2+}$ is required for substrate binding, and $Mn^{2+}$ is favored as the catalytic metal. Highly sensitive web-like protein•NTP, NTP•NTP, and intra-NTP interactions precisely position NTPs and the cognate intermediate to generate $2'-5'/3'-5'$-linked cGAMP (indicated in colored boxes/texts).

were expressed in E. coli BL21-(DE3) cells at 16 °C using 0.3 mM IPTG for 18 hours. Cells were lysed by sonication in the lysis buffer (20 mM Na-phosphate at pH 7.5, 500 mM NaCl, 40 mM imidazole, and 0.5 mM TCEP) and purified by the Ni-NTA chromatography. The fusion protein was cleaved by the TEV protease at 4 °C overnight and the MBP tag was removed by binding them to the Ni-NTA column and amylose column. The untagged protein (flow-through) was further purified by size-exclusion chromatography (Superdex 200 for full-length, Superdex 75 for catalytic domain, Cytiva). Human cGAS protein variants were concentrated and stored in 20 mM Tris-HCl at pH 7.5, 300 mM KCl, and 0.5 mM TCEP. Mouse cGAS protein variants were concentrated in 20 mM Tris-HCl, pH 7.5, 150 mM NaCl, and 0.5 mM TCEP. All purified cGAS proteins were flash-frozen in liquid nitrogen and stored at −80 °C.

## Crystallization
All human cGAS crystals were grown by vapor diffusion in hanging drops at 25 °C in 24-well plates within 3 days. apo-WT-hcGAS$^{cat}$ crystals were obtained by mixing 1 µl of protein (8 mg/ml) with 1 µl of reservoir solution containing 0.1 M Tris pH 8.5, and 8 % (w/v) PEG8000. For co-crystallization of WT-hcGAS$^{cat}$ with nucleotides ATP/GTP or GTP, 8 mg/ml protein was firstly preincubated with 2 mM nucleotides, and then 1 µl of protein -nucleotides mixture was mixed with 1 µl of reservoir solution containing 5 mM magnesium chloride, 0.1 M Tris-HCl pH 8.5, and 8% (w/v) PEG8000.

All mouse cGAS crystals were obtained using the hanging-drop vapor-diffusion method at 4 °C in a 24-well plate within one week. For crystals of the apo-mcGAS$^{cat}$ (WT or QN), 10 mg/ml of protein was mixed with 1.2 mM of 16-bp dsRNA (5'- UCU GUA CAU GUA CAG A-3')

at a ratio of 1:1.2 and incubated on ice for 30 min. 1 µl of the mixture was mixed with 1 µl of reservoir solution containing 0.2 M magnesium acetate tetrahydrate, 0.1 M sodium cacodylate trihydrate (pH 6.5), and 20% (w/v) PEG8000. mcGAS$^{cat}$ was crystallized without dsRNA. For crystallization of mcGAS$^{cat}$•dsDNA, 10 mg/ml of protein (WT or QN) was mixed with 1.2 mM of palindromic 18-bp dsDNA (5'-ATC TGT ACA TGT ACA GAT-3') at the 1:1.2 molar ratio and incubated on ice for 30 min. 1 µl of the protein•DNA solution was then mixed with 1 µl of reservoir solution containing 0.2 M ammonium acetate, 32% (v/v) MPD, and 0.1 M Bis-Tris pH 6.5.

## Crystal preparation (soaking)
The mouse cGAS$^{cat}$ and human cGAS$^{cat}$ crystals were harvested and cryoprotected using the reservoir buffer supplemented with 20% (v/v) ethylene glycol before flash-frozen in liquid nitrogen. For substrate binding or reaction in crystals, mcGAS$^{cat}$ or mcGAS$^{cat}$•dsDNA crystals (WT or QN) were first transferred to the reservoir buffer supplemented with 2 mM NTPs (ATP, GTP, ATP/GTP, ATP/ITP, ITP/GTP ATP/2'-dGTP, 2'-dATP, and 3'd-ATP) plus 5 mM MgCl$_2$ (or MnCl$_2$ when noted)) then incubated at 4 °C for 20 min before flash-frozen in liquid nitrogen without cryoprotection. For tracking the anomalous signal of $Mn^{2+}$, WT-mcGAS$^{cat}$•dsDNA crystals were transferred to the reservoir buffer supplemented with 4 mM ATP, 10 mM MgCl$_2$, and varying MnCl$_2$ (1000, 500, 200, 100, 40, 15, and 0 µM) for 20 min at 4 °C then flash-frozen in liquid nitrogen without cryoprotection.

## Data collection and structure determination
All diffraction data were collected from the National Synchrotron Light Source II (NSLS-II) at Brookhaven at 100 K on beamlines 17-ID-1 (AMX)

or 17-ID-2 (FMX). For X-ray anomalous scattering, the data were collected at the absorption edge of manganese (6539 eV) on 17-ID-2 (FMX). Data were indexed and integrated with XDS[28] and scaled and merged with AIMLESS[29]. All the structures were solved by molecular replacement using the MOLREP module in the CCP4i suite[29]. We used an apo-mcGAS$^{cat}$ monomer (from PDB: 4lez) and the 1:1 mcGAS$^{cat}$•dsDNA complex (without bound cGAMP and the second cGAS monomer and dsDNA in 4lez) as the initial search models for mcGAS$^{cat}$ and mcGAS$^{cat}$•dsDNA, respectively. For hcGAS$^{cat}$, we used PDB ID: 4k8v as the search model. We ensured that all search models have empty active sites. The subsequent model building and structure refinement were conducted in Coot[30] and Phenix_Refine[31]. The Mn$^{2+}$ anomalous maps were calculated using REFMAC5 and quantified in Coot using the contour level (σ). The structure figures were prepared using CCP4MG[29] and PyMOL (Schrödinger). Data collection and refinement statistics are summarized in Supplementary Tables 1–9. The coordinates and structural factors of all structures have been deposited to the Protein Data Bank with the respective PDB IDs listed in Supplementary Tables 1–9.

### Pyrophosphatase-coupled cGAS activity assay

The catalytic activity of cGAS (NTase) was measured by the pyrophosphatase-coupled assay tracking the production of PP$_i$ as reported[15,18,19]. Briefly, 200 nM cGAS was incubated with 50 nM of E. coli pyrophosphatase, 200 μM of NTPs (ATP/GTP, ATP/ITP, PuTP/ GTP, PuTP/ATP, ATP/ATP, GTP/GTP, ITP/ITP, ATP/2'-dGTP, 3'-ATP/ GTP, 2'-dATP/GTP) plus 200 nM 60-bp dsDNA in the reaction buffer (25 mM Tris-acetate (Ac) pH 7.4, 125 mM KAc, 1 mM TCEP, 5 mM MgAc$_2$ at pH 7.4, and 5% glycerol) at 25 ± 2 °C (RT). 60-bp dsDNA sequence: 5'-ATG GAA GAT CCG CGT AGA AGG ACG ACG GCG CCA CGC GCT AAG AAG CCG TCC GCG AAG CGC-3'. A total of 4 aliquots were taken from the reaction at designated time points and mixed with an equal volume of the quench buffer (the reaction buffer plus 25 mM EDTA) in a 384-well plate (Corning). Quenched reactions were mixed with 10 μl malachite green solution and incubated for 45 min at RT. The absorbance at 620 nm was recorded using a Tecan M1000 plate reader, and data were compared to an internal standard curve of inorganic phosphate to determine the concentration of phosphate released in each well. Phosphate concentrations of control reactions without cGAS were subtracted from reactions containing the enzyme. Apparent catalytic rates were calculated from the slopes of control-subtracted phosphate concentrations over time.

### Isothermal titration calorimetry

Experiments were performed using a MicroCal VP-ITC microcalorimeter (Malvern) under the reaction buffer containing 20 mM HEPES at pH 7.5, 125 mM KAc, 5 mM MgCl$_2$, and 1 mM TCEP. For nucleotide-cGAS interactions, 0.8 mM nucleotide in the injection syringe was titrated into the sample cell loaded with 10 μM cGAS protein with or without 10 μM 18-bp dsDNA (5'-ATC TGT ACA TGT ACA GAT-3'). For interactions of nucleotide with cGAS pre-bound with NTPs, 0.8 mM nucleotide in the injection syringe was titrated into the sample cells loaded with 10 μM cGAS and 120 μM nucleotide with or without 10 μM 18-bp dsDNA. Titrations consisted of 20 injections (size 15 μl, duration 18 sec), with 300 sec equilibration time. The data were analyzed using Origin 7.0.

### Analytical fractionation of cGAS reactions by HPLC

1 μM cGAS, 1,000 μM NTPs, and 1 μM 60-bp dsDNA were incubated in 50 μl reaction buffer (25 mM Tris-Ac pH 7.4, 125 mM KAc, 1 mM TCEP, 5 mM MgAc$_2$ at pH 7.4, and 5% glycerol) for the indicated time at 25 ± 2 °C. Reactions were quenched with 25 mM EDTA, diluted to 80 μl with water, and filtered with a 3 kDa cutoff centrifugal filter

(VWR). The flow-through was then mixed 1:1 with HPLC buffer A (see below) and fractionated on the Waters 1525 HPLC system using a Poroshell 120 SB-C18 column (2.7 μm; 4.6 × 100 mm) with a 100 μl sample loop. Absorbance at 260 nm was tracked with a Waters 2996 photodiode array detector. To degrade 3'-5'-linked dinucleotides, reaction samples were first passed through in 3 kDa cutoff centrifugal filters, then treated with the S1 nuclease (2 units/μl final; ThermoFisher) for 15 min at 37 °C. All reactions were then quenched with 25 mM EDTA and filtered again. HPLC gradient scheme: 0% B from 0–1 min, linear increase to 50% B from 1–27 min, and linear increase to 100% B from 27–28 min. HPLC gradient scheme to resolve pppGpG and ATP (related to Fig. 3):0% B from 0–1 min, linear increase to 50% B from 1–37 min, and linear increase to 100% B from 37–38 min. Buffer A: 100 mM potassium phosphate monobasic, 5 mM tetrabutylammonium, final pH 6.0; Buffer B: buffer A supplement with 30% acetonitrile. Peak intensities were integrated by the Waters Empower™ 3 software.

### Fluorescence-anisotropy binding assays

An increasing amount of cGAS was added to fluorescein-amidite (FAM)-labeled dsDNA fragments (5 nM final). Fluorescence anisotropy (FA) signal was recorded with a Tecan M1000 plate reader as previously reported[15,18]. Changes in FA were plotted as a function of cGAS concentration and fit to the Hill equation. 20-bp FAM-dsDNA: 5'-TAA GAC ACG ATG CGA TAA AA-3'. 39-bp FAM-dsDNA: 5'-TAA GAC ACG ATG CGA TAA AAT CTG TTT GTA AAA TTT ATT-3'. 72-bp FAM-dsDNA: 5'-TAA GAC ACG ATG CGA TAA AAT CTG TTT GTA AAA TTT ATT AAG GGT ACA AAT TGC CCT AGC ACA GGG GTG GGG-3'.

### Measuring cGAS signaling activities in HEK293T cells using the luciferase reporter

The Human Embryonic Kidney (HEK) 293T cells (ATCC, CRL-11268) were maintained in the DMEM high glucose medium (Thermo-Fisher) supplemented with 10% FBS at 37 °C and 5% CO$_2$. Cells were seeded in a 24-well plate (6 ×10$^4$ cells/well) and incubated overnight. 50 ng of pCMV plasmids encoding empty vector or various cGAS$^{FL}$ variants were then transfected using Lipofectamine 3000 (Invitrogen), along with 5 ng of Renilla Luciferase plasmid, 10 ng of plasmid encoding human STING, and 10 ng of plasmid encoding Firefly Luciferase under an IFN-Iβ promoter. Transfected cells were incubated for 24 hours, washed once with PBS, and lysed with passive lysis buffer (Promega). Lysates were transferred to a white 96-well plate with a reflective flat bottom (Corning) and analyzed for Firefly Luciferase and Renilla Luciferase activities using the Dual-Luciferase Reporter Assay System (Promega) on a Synergy H1 plate reader equipped with the dual-injector system (BioTek). The firefly output was divided by the Renilla output and normalized to the empty vector control; see also ref. 18.

### Statistics and reproducibility

All biochemical experiments were conducted at least three times. No data were excluded from the analyses. *p*-values were determined using a 2-tailed *t*-test with equal variances.

### Reporting summary

Further information on research design is available in the Nature Portfolio Reporting Summary linked to this article.

## Data availability

The structure coordinates generated in this study have been deposited in the Protein Data Bank under the following accession codes: 8SHU (apo mcGAS WT). 8SHK (mcGAS WT + ATP). 8SHY (mcGAS QN + ATP). 8SHZ (apo hcGAS WT). 8SI0 (hcGAS WT + cGAMP). 8SJ8 (hcGAS WT + pppGpG). 7UUX (cGAS QN:dsDNA + ATP). 8SKT (cGAS

WT:dsDNA + ATP/Mn$^{2+}$). 7UYQ (cGAS QN:dsDNA + GTP). 7UXW (cGAS QN:dsDNA + ATP/GTP). 8G1J (cGAS QN:dsDNA + ATP/ITP). 8G1O (cGAS QN:dsDNA + GTP/ITP). 7V0W (cGAS QN:dsDNA + pppGpA/Mn$^{2+}$). 7UYZ (cGAS WT:dsDNA + pppGpG/Mg$^{2+}$). 7UZR (cGAS WT:dsDNA + pppGpG/Mn$^{2+}$). 8EAE (cGAS WT:dsDNA + pppGpI). 8G23 (cGAS WT:dsDNA + pppIpA). 8SJ0 (cGAS:dsDNA + 2′-dATP). 8SJ1 (cGAS:dsDNA + 3′-dATP). 8SJ2 (cGAS:dsDNA + 2′-dGTP). 8GIM (cGAS:dsDNA + ATP·10 mM Mg$^{2+}$). 8GIN (cGAS:dsDNA + ATP·10 mM Mg$^{2+}$ + 15 μM Mn$^{2+}$). 8GIP (cGAS:dsDNA + ATP·10 mM Mg$^{2+}$ + 40 μM Mn$^{2+}$). 8GIO (cGAS:dsDNA + ATP·10 mM Mg$^{2+}$ + 100 μM Mn$^{2+}$). 8GIR (cGAS:dsDNA + ATP·10 mM Mg$^{2+}$ + 200 μM Mn$^{2+}$). 8GIS (cGAS:dsDNA + ATP·10 mM Mg$^{2+}$ + 500 μM Mn$^{2+}$). 8GIT (cGAS:dsDNA + ATP·10 mM Mg$^{2+}$ + 1000 μM Mn$^{2+}$). All quantitative data generated in this study are provided in the Supplementary Information file and the Source Data. Source data is provided as a Source Data file. Source data are provided with this paper.

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

## Acknowledgements

We thank Dr. Mario A. Bianchet and the beamline staff for remote crystallographic data collection at NSLS-II at Brookhaven, NY. This work was supported by NIH grants (RO1 GM129342 and R35 GM145363 to J.S.). Work at the AMX (17-ID-1) and FMX (17-ID-2) beamlines was supported by the NIH, the National Institute of General Medical Sciences (P41GM111244), the DOE Office of Biological and Environmental Research (KP1605010), and the National Synchrotron Light Source II at Brookhaven National Laboratory are supported by the DOE Office of Basic Energy Sciences under contract number DE-SC0012704 (KC0401040). In memory of Dr. L. Mario Amzel.

## Author contributions

S.W. and J.S. conceptualized the project. S.W. performed all experiments, analyzed data, and wrote the paper. S.B.G. supervised the crystallographic data collection and processing and also edited the paper. J.S. supervised the overall project, analyzed data, and wrote the paper.

## Competing interests

The authors declare no competing interests.
