## [Peer Review File · Nature Communications]

The structural basis for 2'-5'/3'-5'-cGAMP synthesis by cGASReviewer #1 (Remarks to the Author):

This study presents a series of crystal structures of cGAS in various stages of binding ATP and GTP and a state toward forming the cGAMP product. Despite many structures of cGAS already solved, this set of high-resolution structures addresses the remaining questions regarding the specificity of cGAS for the NTPs and the role of the metal ions in catalysis. The structures are nicely complemented by rigorous measurements of the thermodynamic constants of the NTP binding steps providing the driving forces for the reaction steps. Overall, this is a well-done study that has carefully analyzed the reaction pathway of cGAS using powerful crystallography methods and thermodynamic measurements.

Given that there are many structures of cGAS, the authors need to properly compare their structure with the published ones for the benefit of the readers. How is the apo structure reported here different from the one published? The authors say there is an ensemble of inactive conformations related to the apo structure. How is this assessed through crystallography, where dynamic information is not available? If the apo state has basal activity, why is it referred to as inactive? Also, referring to this state as non-productive is not appropriate.

The authors refer to a basal activity, but I could not find the basal activity measured in any of the figures. How much stimulation does the DNA provide?

Specificity for ATP and GTP at sites 1 and 2 - I am struggling to understand what interactions provide the specificity for ATP and GTP that is succinctly described in the results. Panel 3c shows the base in ATP stacking with Y421 and near D371. Are these providing specificity for ATP? This is not discussed clearly in the text. The authors conclude that "the -NH₂ of ATP contributes positively to the recognition of the ATP/GTP pair." This is a bit vague.

line 248 - "Our observations strongly suggest that GTP binding and the first 2'-5' signature bond formation are tightly coupled" should be justified better.

Does Mn²⁺ increase the basal activity of cGAS?

Fig 5B panel is difficult to comprehend. Why is the ATP of pppGpA compared to GTP and not ATP? Also, the GTP can be made gray to distinguish it from the pppGpA.

Reviewer #2 (Remarks to the Author):

This manuscript by Wu et al provides a detailed analyses of the catalytic mechanism of cGAS, the cytoplasmic double-stranded DNA sensor that catalyzes the production of the second messenger cGAMP as an innate immunity trigger. The study presents a host of crystal structures of cGAS in various states with different cognate or non-cognate substrates bound, revealing the details of the interactions between the catalytic site residues and the substrates or products. Together with the binding measurements and enzymatic activity assays, the structures help delineate the key interactions in determining the substrate specificity as well as the specific bonds formed in the product (2'-5' versus 3'-5' linkage, which are critical factors that can determine different downstream immune responses). The work is well done and the data presented appear very solid, although many points presented represent a collection of detailed features of the cGAS enzyme that either confirm or modify slightly what are already known. There are several points that of greater interest and novelty. For example, based on the structures, the authors propose that two residues interacting with the adenosine base at site 2 dictate the specific angle of the base and the ribose group, which in turn specifies the 3'-5' bond formation in the second reaction step. The comparison of mouse and human cGAS suggests that differences at a few residues could lead to different enzymatic kinetics and signaling.

Specific points:

Lines 157-160. These statements do not seem well justified. Lack of effects of DsDNA on cGAMP binding do not necessary suggest that the product is rapidly released. How do the binding measurements show that "dsDNA binding affixes Mg²⁺/NTPs to the catalytically competent conformation"?

Lines 219-226. The manuscript states that the densities clearly show which nucleotide is bound site I or II. However, the densities shown in Figure 3E and F (and in the PDB validation reports) do not look very clear, and it is not obvious how the authors reached the assignment as shown, given that the shapes of the bases are highly similar.

Line 318-320, I am confused by the statement regarding the role of the -NH₂ of guanine in affixing the metal/intermediate complex to the catalytically competent state, which seems out of context.

Figure 6F shows that different variants of mouse and human cGAS induce similar levels of INF responses, leading to the conclusion (in the result and discussion) that the catalytic activity is not directly correlated with the signaling outcome. This seems an oversimplified conclusion given that the experimental system is artificial (human cells with transfected cGAS and STING from different species), which may fail to capture small but functionally significant differences between signals triggered by human and mouse cGAS.

Minor points:

The spheres of metal ions in figures are too big and obscure the sidechains of surrounding residues. It might be better to reduce the size of the spheres.

The color scheme of the labels and structures in some cases is difficult to discern. For example, in Figure 5C, it is hard to tell which label is correspond to which structure unless one zooms in extensively to see the edge color of the labels. It might be better to color the entire label instead of just the edge.

It seems that Figure 7 is not referred to in the text.

Reviewer #3 (Remarks to the Author):

Shortly after the discovery of cGAS about a decade ago, several groups solved the structures of cGAS in its apo form and DNA-bound form, which revealed a DNA-induced conformational change in the active site of cGAS that underlies the activation of its enzymatic activity, which catalyzes the conversion of GTP and ATP into a cGAMP molecule that contains both 2'-5' and 3'-5' phosphodiester bonds (2'3'-cGAMP). Among these published papers, two also showed that cGAS forms a dimer that binds to two molecules of DNA (2 cGAS:2 DNA) and that this dimerization of cGAS is important for its activation. Subsequent studies also showed that the binding of long dsDNA to cGAS induces liquid-liquid phase separation, which is important for its activation. In this new study, Wu et al solved several crystal structures of cGAS at different stages of its activation process, which provides a more detailed insight into the catalytic mechanism, including the specificity of substrate binding and the formation of specific phosphodiester linkages[(G(2'5')pA(3'5')p). While it is commendable to undertake this effort to elucidate a more detailed mechanism of cGAS catalysis, the major conclusions from this paper largely confirm the previous findings, especially regarding DNA-induced alignment of the catalytic residues in the active site of the enzyme. This paper would be more impactful if it could provide more insights into some remaining questions about cGAS regulation and catalysis. Some comments are elaborated below:

1) As a cyclase, cGAS catalyzes the formation of two phosphodiester bonds in a sequential manner using the same catalytic residues in the active site. It's remarkable that after the formation of the linear di-nucleotide (pppGpA), this reaction intermediate is reoriented in the active site in such a

way that enables the formation of the second diester bond, resulting the formation of the cyclic dinucleotide cGAMP. How is the formation of the first phosphodiester bond coupled to the subsequent 'flipping' or reorientation of pppGpA and the formation of the second phosphodiester bond? Some understanding of the kinetics including the rate-limiting step and conformational dynamics in this multi-step process would be needed to add to the understanding of cGAS catalysis.

2) The authors stated in the abstract that "dimerization-coupled dsDNA-binding then affixes the active site into a rigid lock for productive metal-substrate binding". While it's now well accepted that dimerization of cGAS is important for its activation and that DNA binding induces a conformational change in the active site of cGAS, how dimerization plays a role in cGAS catalytic activation is not as well understood. Does the authors' structural work provide a mechanistic explanation for the requirement of cGAS dimerization in its activation?

3) The authors emphasize the importance of Mn^{2+} in cGAS activation. However, whether Mn^{2+} plays a role in cGAS activation in vivo remains uncertain because Mg^{2+} is sufficient to support cGAS catalysis in vitro and the physiological concentration of Mg^{2+} is much higher than Mn^{2+} in the cell's cytoplasm.

We'd like to thank the reviewers for their insightful comments. Please find our replies below. We also highlighted any changes we made in the manuscript.

Reviewer 1

1. Given that there are many structures of cGAS, the authors need to properly compare their structure with the published ones for the benefit of the readers. How is the apo structure reported here different from the one published?

In addition to Fig. 1B and 1D, we made a new Fig. S1A displaying our two apo-structures aligned to previously reported apo-structures. As shown in these figures, apo-cGAS^{cat} structures show multiple conformations for both the beta-sheets harboring the active site residues and the activation loop.

2. The authors say there is an ensemble of inactive conformations related to the apo structure. How is this assessed through crystallography, where dynamic information is not available?

We intended to state that apo-cGAS likely assumes multiple different conformations based on (1) the high B-factors at the active site without dsDNA (Fig S1D), and (2) our structural models showing different active site conformations from those published (Figs 1 and S1). We do not claim that our structure is right and the previously reported structures are wrong. Instead, these observations indicate that cGAS can assume many different conformations.

We changed “ensemble” to “array.”

If the apo state has basal activity, why is it referred to as inactive? Also, referring to this state as non-productive is not appropriate. The authors refer to a basal activity, but I could not find the basal activity measured in any of the figures. How much stimulation does the DNA provide?

We previously looked at the dsDNA-dependent activation of cGAS (Hooy et al., *eLife*, 2018). Without dsDNA, cGAS has basal activity of $\sim 150 \text{ M}^{-1}\text{min}^{-1}$, which is amplified by dsDNA binding by 40-fold. We updated the citation in line 112.

Specificity for ATP and GTP at sites 1 and 2 - I am struggling to understand what interactions provide the specificity for ATP and GTP that is succinctly described in the results. Panel 3c shows the base in ATP stacking with Y421 and near D371. Are these providing specificity for ATP? This is not discussed clearly in the text. The authors conclude that " the -NH2 of ATP contributes positively to the recognition of the ATP/GTP pair." This is a bit vague.

When NTPs bind at site-1, our structure (Fig 3C) shows that E371 stabilizes binding by forming H-bonds with the 2'-OH. It appears that Y421 is simply important for base-stacking against purines regardless of A vs. G (Gao et al., *Cell* 2013); indeed, at site-1, both ATP and GTP bind similarly. (Fig. 3D).

When we soaked the QN mutant with either ATP alone or GTP alone, we found ATP only at site-1 while GTP binds to both site-1 and site-2 (e.g., Fig. S2-B/D). These observations suggest that site-2 intrinsically favors G over A. Moreover, in our ATP/GTP-bound structure (Fig. 3D), we noted that the NH₂ of ATP at site-1 appears to donate an H-bond to the α-phosphate of GTP at site-2; however, such an inter-NTP interaction is unlikely when GTP binds at site-1, as the carboxyl oxygen of G is a H-bond acceptor. These observations support the idea that the -NH₂ of ATP contributes positively to the recognition of the ATP/GTP pair. Furthermore, our biochemical assays show that cGAS does not utilize PuTP (ATP *sans* -NH₂) as efficiently as ATP (Fig. 3G)

line 248 - "Our observations strongly suggest that GTP binding and the first 2'-5' signature bond formation are tightly coupled" should be justified better.

Thanks for the comment. We now realize that such a statement is not necessary at this point, as we expand this idea further in the next paragraph. We deleted this sentence.

Does Mn²⁺ increase the basal activity of cGAS?

It was previously reported that cytosolic Mn²⁺ boosts the signaling activity of cGAS (Wu et al., 2018, *Immunity*). We (Hooy et al, *NAR*, 2020) and others (Zhao et al., *Cell Reports*, 2020) found that Mn²⁺ can activate cGAS without dsDNA, albeit requiring artificially high concentrations. In our previous study, we found that Mn²⁺ is more readily incorporated over Mg²⁺ at physiologically relevant concentrations in the dsDNA-dependent activation of cGAS (Hooy et al, *NAR*, 2020). In this manuscript, we show that Mn²⁺ is preferentially utilized as the catalytic metal, again at physiological concentrations.

Fig 5B panel is difficult to comprehend. Why is the ATP of pppGpA compared to GTP and not ATP? Also, the GTP can be made gray to distinguish it from the pppGpA.

Unlike the A portion of the pppGpA dinucleotide, ATP fails to bind stably at site-2 in all known structures. It is likely that triphosphate of pppGpA anchors the intermediate. We compared the A portion of pppGpA to GTP to demonstrate how G and A bind differently at site-2 to prescribe the 2'-5'-linkage vs. the 3'-5'-linkage, respectively. We changed GTP color to dark gray in Figure 5B.

Reviewer 2

Lines 157-160. These statements do not seem well justified. Lack of effects of DsDNA on cGAMP binding do not necessary suggest that the product is rapidly released.

Thanks for the comment. We changed the statement- “Also of note, not only was cGAMP binding barely affected by dsDNA, but it also bound more weakly than ATP/GTP, suggesting that the product would not interfere with catalysis.”

How do the binding measurements show that “dsDNA binding affixes Mg²⁺/NTPs to the catalytically competent conformation”?

We made the above statement because dsDNA binding resulted in more favorable enthalpy (consistent with bond formation; Fig. 2C) and unfavorable entropy (consistent with reduction in degree of freedom; Fig. 2D) for binding Mg²⁺•NTP. These are consistent with our crystal structures.

Lines 219-226. The manuscript states that the densities clearly show which nucleotide is bound site I or II. However, the densities shown in Figure 3E and F (and in the PDB validation reports) do not look very clear, and it is not obvious how the authors reached the assignment as shown, given that the shapes of the bases are highly similar.

We thought long and hard about this and believe that multiple lines of evidence support our stated conclusions:

In the crystal structure of GTP/ITP-bound QN-mcGAS^{cat}•dsDNA (Fig. 3E), the electron density map indicates that the NTP bound at site-2 has -NH₂ while the NTP at site-1 doesn't. Our observation suggests that GTP and ITP are bound at site-2 and site-1, respectively. Moreover, the GTP/ITP produced the pppGpI dinucleotide when soaked into WT-mcGAS^{cat}•dsDNA crystal, as evidenced by the presence and absence of -NH₂ at site-1 and site-2, respectively (Figs. S5B and S8-2); i.e., GTP was bound at site-2 and ITP was bound at site-1 for dinucleotide formation. The increased NTase rate (Fig. 3G and Table S2) for ITP/ITP over GTP/GTP also support that cGAS prefers ITP at site-1 (i.e., the -NH₂ of GTP is disfavored at site-1).

Similarly, in the crystal structure of ATP/ITP-bound QN-mcGAS^{cat}•dsDNA (Fig. 3F), we believe ITP is bound at site-2 based on its resemblance to how GTP is bound here via the H-bond between its carboxyl oxygen and Ser³⁶⁶ (e.g., Figs. 3E and S2A-B). Of note, adenine interacts with Ser³⁶⁶ via its N1 imine in the pppGpA intermediate (Fig. 5) and ATP fails to stably bind at site-2 in all our structures (e.g., Fig. S2D). Moreover, ATP/ITP produces dinucleotide pppIpA in WT-mcGAS^{cat}•dsDNA crystal, as the A portion of the pppIpA dinucleotide highly resembles pppGpA (Fig. 5D), but not GTP (Fig. 5B). Additionally, cGAS utilizes ATP/ITP better than GTP/GTP or ITP/ITP, supporting that ATP (via its -NH₂) is favored at site-1 (Fig. 3G and Table S2).

Line 318-320, I am confused by the statement regarding the role of the -NH₂ of guanine in affixing the metal/intermediate complex to the catalytically competent state, which seems out of context.

ITP lacks the -NH₂ and the reaction is apparently stuck at the dinucleotide pppIpA stage without cyclization (Fig. 5D and Fig. S4). We do not see the second catalytic metal in our pppIpA-bound structure (Fig. 5D). By contrast, in the pppGpA-bound structure, the -NH₂ is apparently making an H-bond with the α-phosphate.

To clarify, we changed the sentence: “...-NH₂ of guanine is important for positioning the metal•intermediate complex for cyclization.”

Figure 6F shows that different variants of mouse and human cGAS induce similar levels of INF responses, leading to the conclusion (in the result and discussion) that the catalytic activity is not directly correlated with the signaling outcome. This seems an oversimplified conclusion given that the experimental system is artificial (human cells with transfected cGAS and STING from different species), which may fail to capture small but functionally significant differences between signals triggered by human and mouse cGAS.

We used human STING in both experiments. To clarify this, we added “human” in Figure legend 6F. We also added in the discussion: “It is still possible that there might be small but functional differences that our transfection system failed to capture.”

The spheres of metal ions in figures are too big and obscure the sidechains of surrounding residues. It might be better to reduce the size of the spheres.

Thanks for the suggestion. We considered this and used smaller spheres in supplementary figures that show electron density maps. We think having these two different representations would help readers.

The color scheme of the labels and structures in some cases is difficult to discern. For example, in Figure 5C, it is hard to tell which label is correspond to which structure unless one zooms in extensively to see the edge color of the labels. It might be better to color the entire label instead of just the edge.

Thanks for the comment. We changed the coloring scheme in Fig 5.

It seems that Figure 7 is not referred to in the text.

We refer to Figure 7 in line 369 and multiple places in the Discussion.

Reviewer 3

As a cyclase, cGAS catalyzes the formation of two phosphodiester bonds in a sequential manner using the same catalytic residues in the active site. It's remarkable that after the formation of the linear di-nucleotide (pppGpA), this reaction intermediate is reoriented in the active site in such a way that enables the formation of the second diester bond, resulting the formation of the cyclic dinucleotide cGAMP. How is the formation of the first phosphodiester bond coupled to the subsequent 'flipping' or reorientation of pppGpA and the formation of the second phosphodiester bond? Some understanding of the kinetics including the rate-limiting step and conformational dynamics in this multi-step process would be needed to add to the understanding of cGAS catalysis.

We greatly appreciate this insightful comment. Time-resolving cGAS reaction with WT-mcGAS^{cat}•dsDNA crystals soaked with ATP/GTP/Mg²⁺ was one of the directions we pursued in our crystallographic studies. We have some new findings that are in preparation. We are able to trap the ppp(2'-5')GpA in the cGAS reaction and generate a large quantity of ppp(2'-5')GpA. With that, we hope to measure the kinetics of the rate-limiting step (2'-5' linkage vs. 3'-5' linkage). We are also working on further probing this question through Molecular Dynamics (MD) simulations. We look forward to reporting these findings in the near future.

The authors stated in the abstract that “dimerization-coupled dsDNA-binding then affixes the active site into a rigid lock for productive metal-substrate binding”. While it’s now well accepted that dimerization of cGAS is important for its activation and that DNA binding induces a conformational change in the active site of cGAS, how dimerization plays a role in cGAS catalytic activation is not as well understood. Does the authors’ structural work provide a mechanistic explanation for the requirement of cGAS dimerization in its activation?

In our ITC experiments, we found that the binding of substrate (ATP or GTP) to the dimerization-deficient mutant (K382E) is insensitive to dsDNA-binding, and this mutant does not display any preference for the ATP-GTP pair (Figure 4B). Combined with our previous observation that dimerization is coupled to dsDNA length-dependent binding (Hooy et al., *eLife*, 2018), our studies indicate that dimerization is integral to dsDNA-dependent binding of the cognate substrate pair.

The authors emphasize the importance of Mn²⁺ in cGAS activation. However, whether Mn²⁺ plays a role in cGAS activation in vivo remains uncertain because Mg²⁺ is sufficient to support cGAS catalysis in vitro and the physiological concentration of Mg²⁺ is much higher than Mn²⁺ in the cell’s cytoplasm.

It was previously reported that cytosolic Mn²⁺ boosts the signaling activity of cGAS (Wu et al., 2018, *Immunity*). We (Hooy et al, *NAR*, 2020) and others (Zhao et al., *Cell Reports*, 2020) found that Mn²⁺ can activate cGAS without dsDNA, albeit requiring artificially high concentrations. In our previous study, we found that Mn²⁺ is more readily incorporated over Mg²⁺ at physiologically relevant concentrations in the dsDNA-dependent activation of cGAS (Hooy et al, *NAR*, 2020). In the current manuscript, we find that Mn²⁺ is preferentially incorporated as the catalytic metal even in the presence of excess Mg²⁺ (10 mM).

Reviewer #1 (Remarks to the Author):

The authors have addressed my concerns.

Reviewer #2 (Remarks to the Author):

The authors addressed my previous questions adequately in the rebuttal letter. However, they made minimal changes to the manuscript. I believe that some of the answers provided in the rebuttal letter to me and the other two reviews should be incorporated into the manuscript. Doing so would improve the clarity of the paper.

We incorporated our responses to the manuscript as suggested. We marked reviewers' comments/our responses and the corresponding edits in the manuscript by highlighting them with the same colors.

Reviewer 1

1. Given that there are many structures of cGAS, the authors need to properly compare their structure with the published ones for the benefit of the readers. How is the apo structure reported here different from the one published?

In addition to Fig. 1B and 1D, we made a new Fig. S1A displaying our two *apo*-structures aligned to previously reported *apo*-structures. As shown in these figures, *apo*-cGAS^{cat} structures show multiple conformations for both the beta-sheets harboring the active site residues and the activation loop. ***already incorporated**

2. The authors say there is an ensemble of inactive conformations related to the apo structure. How is this assessed through crystallography, where dynamic information is not available?

We intended to state that *apo*-cGAS likely assumes multiple different conformations based on (1) the high B-factors at the active site without dsDNA (Fig S1D), and (2) our structural models showing different active site conformations from those published (Figs 1 and S1). We do not claim that our structure is right and the previously reported structures are wrong. Instead, these observations indicate that cGAS can assume many different conformations.

We changed “ensemble” to “array.” ***already incorporated**

If the apo state has basal activity, why is it referred to as inactive? Also, referring to this state as non-productive is not appropriate. The authors refer to a basal activity, but I could not find the basal activity measured in any of the figures. How much stimulation does the DNA provide?

We previously looked at the dsDNA-dependent activation of cGAS (Hooy et al., *eLife*, 2018). Without dsDNA, cGAS has basal activity of $\sim 150 \text{ M}^{-1}\text{min}^{-1}$, which is amplified by dsDNA binding by 40-fold. We updated the citation in line 112. ***already incorporated**

Specificity for ATP and GTP at sites 1 and 2 - I am struggling to understand what interactions provide the specificity for ATP and GTP that is succinctly described in the results. Panel 3c shows the base in ATP stacking with Y421 and near D371. Are these providing specificity for ATP? This is not discussed clearly in the text. The authors conclude that " the -NH2 of ATP contributes positively to the recognition of the ATP/GTP pair." This is a bit vague.

**** please note that this point is similar to what Reviewer 2 indicated, we thus highlighted them in with the same color****

When NTPs bind at site-1, our structure (Fig 3C) shows that E371 stabilizes binding by forming H-bonds with the 2'-OH. It appears that Y421 is simply important for base-stacking against purines regardless of A vs. G (Gao et al., *Cell* 2013); indeed, at site-1, both ATP and GTP bind similarly. (Fig. 3D).

When we soaked the QN mutant with either ATP alone or GTP alone, we found ATP only at site-1 while GTP binds to both site-1 and site-2 (e.g., Fig. S2-B/D). These observations suggest that site-2 intrinsically favors G over A. Moreover, in our ATP/GTP-bound structure (Fig. 3D), we noted that the NH₂ of ATP at site-1 appears to donate an H-bond to the α-phosphate of GTP at site-2; however, such an inter-NTP interaction is unlikely when GTP binds at site-1, as the carboxyl oxygen of G is a H-bond acceptor. These observations support the idea that the -NH₂ of ATP contributes positively to the recognition of the ATP/GTP pair. Furthermore, our biochemical assays show that cGAS does not utilize PuTP (ATP *sans* -NH₂) as efficiently as ATP (Fig. 3G)

line 248 - "Our observations strongly suggest that GTP binding and the first 2'-5' signature bond formation are tightly coupled" should be justified better.

Thanks for the comment. We now realize that such a statement is not necessary at this point, as we expand this idea further in the next paragraph. We deleted this sentence. *already incorporated

Does Mn²⁺ increase the basal activity of cGAS?

** please note that this point is similar to what Reviewer 3 indicated, we thus highlighted them in with the same color***

It was previously reported that cytosolic Mn²⁺ boosts the signaling activity of cGAS (Wu et al., 2018, *Immunity*). We (Hooy et al, *NAR*, 2020) and others (Zhao et al., *Cell Reports*, 2020) found that Mn²⁺ can activate cGAS without dsDNA, albeit requiring artificially high concentrations. In our previous study, we found that Mn²⁺ is more readily incorporated over Mg²⁺ at physiologically relevant concentrations in the dsDNA-dependent activation of cGAS (Hooy et al, *NAR*, 2020). In this manuscript, we show that Mn²⁺ is preferentially utilized as the catalytic metal, again at physiological concentrations.

Fig 5B panel is difficult to comprehend. Why is the ATP of pppGpA compared to GTP and not ATP? Also, the GTP can be made gray to distinguish it from the pppGpA.

Unlike the A portion of the pppGpA dinucleotide, ATP fails to bind stably at site-2 in all known structures. It is likely that triphosphate of pppGpA anchors the intermediate. We compared the A portion of pppGpA to GTP to demonstrate how G and A bind differently at site-2 to prescribe the 2'-5'-linkage vs. the 3'-5'-linkage, respectively. We changed GTP color to dark gray in Figure 5B. *already incorporated

Reviewer 2

Lines 157-160. These statements do not seem well justified. Lack of effects of dsDNA on cGAMP binding do not necessary suggest that the product is rapidly released.

Thanks for the comment. We changed the statement- “Also of note, not only was cGAMP binding barely affected by dsDNA, but it also bound more weakly than ATP/GTP, suggesting that the product would not interfere with catalysis.”

How do the binding measurements show that “dsDNA binding affixes Mg²⁺/NTPs to the catalytically competent conformation”?

We made the above statement because dsDNA binding resulted in more favorable enthalpy (consistent with bond formation; Fig. 2C) and unfavorable entropy (consistent with reduction in degree of freedom; Fig. 2D) for binding Mg²⁺•NTP. These are consistent with our crystal structures.

Lines 219-226. The manuscript states that the densities clearly show which nucleotide is bound site I or II. However, the densities shown in Figure 3E and F (and in the PDB validation reports) do not look very clear, and it is not obvious how the authors reached the assignment as shown, given that the shapes of the bases are highly similar.

**** please note that this point is similar to what Reviewer 1 indicated, we thus highlighted them with the same color****

We thought long and hard about this and believe that multiple lines of evidence support our stated conclusions:

In the crystal structure of GTP/ITP-bound QN-mcGAS^{cat}•dsDNA (Fig. 3E), the electron density map indicates that the NTP bound at site-2 has -NH₂ while the NTP at site-1 doesn't. Our observation suggests that GTP and ITP are bound at site-2 and site-1, respectively. Moreover, the GTP/ITP produced the pppGpI dinucleotide when soaked into WT-mcGAS^{cat}•dsDNA crystal, as evidenced by the presence and absence of -NH₂ at site-1 and site-2, respectively (Figs. S5B and S8-2); i.e., GTP was bound at site-2 and ITP was bound at site-1 for dinucleotide formation. The increased NTase rate (Fig. 3G and Table S2) for ITP/ITP over GTP/GTP also support that cGAS prefers ITP at site-1 (i.e., the -NH₂ of GTP is disfavored at site-1).

Similarly, in the crystal structure of ATP/ITP-bound QN-mcGAS^{cat}•dsDNA (Fig. 3F), we believe ITP is bound at site-2 based on its resemblance to how GTP is bound here via the H-bond between its carboxyl oxygen and Ser³⁶⁶ (e.g., Figs. 3E and S2A-B). Of note, adenine interacts with Ser³⁶⁶ via its N1 imine in the pppGpA intermediate (Fig. 5) and ATP fails to stably bind at site-2 in all our structures (e.g., Fig. S2D). Moreover, ATP/ITP produces dinucleotide pppIpA in WT-mcGAS^{cat}•dsDNA crystal, as the A portion of the pppIpA dinucleotide highly resembles pppGpA (Fig. 5D), but not GTP (Fig. 5B). Additionally, cGAS utilizes ATP/ITP better than

GTP/GTP or ITP/ITP, supporting that ATP (via its -NH₂) is favored at site-1 (Fig. 3G and Table S2).

Line 318-320, I am confused by the statement regarding the role of the -NH₂ of guanine in affixing the metal/intermediate complex to the catalytically competent state, which seems out of context.

ITP lacks the -NH₂ and the reaction is apparently stuck at the dinucleotide pppIpA stage without cyclization (Fig. 5D and Fig. S4). We do not see the second catalytic metal in our pppIpA-bound structure (Fig. 5D). By contrast, in the pppGpA-bound structure, the -NH₂ is apparently making an H-bond with the α -phosphate.

To clarify, we changed the sentence: “...-NH₂ of guanine is important for positioning the metal•intermediate complex for cyclization.” *already incorporated

Figure 6F shows that different variants of mouse and human cGAS induce similar levels of INF responses, leading to the conclusion (in the result and discussion) that the catalytic activity is not directly correlated with the signaling outcome. This seems an oversimplified conclusion given that the experimental system is artificial (human cells with transfected cGAS and STING from different species), which may fail to capture small but functionally significant differences between signals triggered by human and mouse cGAS.

We used human STING in both experiments. To clarify this, we added “human” in Figure legend 6F. We also added in the discussion: “It is still possible that there might be small but functional differences that our transfection system failed to capture.” *already incorporated

The spheres of metal ions in figures are too big and obscure the sidechains of surrounding residues. It might be better to reduce the size of the spheres.

Thanks for the suggestion. We considered this and used smaller spheres in supplementary figures that show electron density maps. We think having these two different representations would help readers. *already incorporated

The color scheme of the labels and structures in some cases is difficult to discern. For example, in Figure 5C, it is hard to tell which label is correspond to which structure unless one zooms in extensively to see the edge color of the labels. It might be better to color the entire label instead of just the edge.

Thanks for the comment. We changed the coloring scheme in Fig 5. *already incorporated

It seems that Figure 7 is not referred to in the text.

We refer to Figure 7 in line 369 and multiple places in the Discussion. *already incorporated

Reviewer 3

As a cyclase, cGAS catalyzes the formation of two phosphodiester bonds in a sequential manner using the same catalytic residues in the active site. It's remarkable that after the formation of the linear di-nucleotide (pppGpA), this reaction intermediate is reoriented in the active site in such a way that enables the formation of the second diester bond, resulting the formation of the cyclic dinucleotide cGAMP. How is the formation of the first phosphodiester bond coupled to the subsequent 'flipping' or reorientation of pppGpA and the formation of the second phosphodiester bond? Some understanding of the kinetics including the rate-limiting step and conformational dynamics in this multi-step process would be needed to add to the understanding of cGAS catalysis.

We greatly appreciate this insightful comment. Time-resolving cGAS reaction with WT-mcGAS^{cat}•dsDNA crystals soaked with ATP/GTP/Mg²⁺ was one of the directions we pursued in our crystallographic studies. We have some new findings that are in preparation. We are able to trap the ppp(2'-5')GpA in the cGAS reaction and generate a large quantity of ppp(2'-5')GpA. With that, we hope to measure the kinetics of the rate-limiting step (2'-5' linkage vs. 3'-5' linkage). We are also working on further probing this question through Molecular Dynamics (MD) simulations. We look forward to reporting these findings in the near future.

The authors stated in the abstract that “dimerization-coupled dsDNA-binding then affixes the active site into a rigid lock for productive metal-substrate binding”. While it's now well accepted that dimerization of cGAS is important for its activation and that DNA binding induces a conformational change in the active site of cGAS, how dimerization plays a role in cGAS catalytic activation is not as well understood. Does the authors' structural work provide a mechanistic explanation for the requirement of cGAS dimerization in its activation?

In our ITC experiments, we found that the binding of substrate (ATP or GTP) to the dimerization-deficient mutant (K382E) is insensitive to dsDNA-binding, and this mutant does not display any preference for the ATP-GTP pair (Figure 4B). Combined with our previous observation that dimerization is coupled to dsDNA length-dependent binding (Hooy et al., *eLife*, 2018), our studies indicate that dimerization is integral to dsDNA-dependent binding of the cognate substrate pair.

The authors emphasize the importance of Mn²⁺ in cGAS activation. However, whether Mn²⁺ plays a role in cGAS activation in vivo remains uncertain because Mg²⁺ is sufficient to support cGAS catalysis in vitro and the physiological concentration of Mg²⁺ is much higher than Mn²⁺ in the cell's cytoplasm.

**** please note that this point is similar to what Reviewer 1 indicated, we thus highlighted them in with the same color****

It was previously reported that cytosolic Mn²⁺ boosts the signaling activity of cGAS (Wu et al., 2018, *Immunity*). We (Hooy et al, *NAR*, 2020) and others (Zhao et al., *Cell Reports*, 2020) found

that Mn^{2+} can activate cGAS without dsDNA, albeit requiring artificially high concentrations. In our previous study, we found that Mn^{2+} is more readily incorporated over Mg^{2+} at physiologically relevant concentrations in the dsDNA-dependent activation of cGAS (Hooy et al, NAR, 2020). In the current manuscript, we find that Mn^{2+} is preferentially incorporated as the catalytic metal even in the presence of excess Mg^{2+} (10 mM).